# Neurotoxin-mediated potent activation of the axon degeneration regulator SARM1

Andrea Loreto[1]*[†], Carlo Angeletti[2†], Weixi Gu[3], Andrew Osborne[1§], Bart Nieuwenhuis[1#], Jonathan Gilley[1], Elisa Merlini[1], Peter Arthur-Farraj[1], Adolfo Amici[2], Zhenyao Luo[3], Lauren Hartley-Tassell[4], Thomas Ve[4], Laura M Desrochers[5¶], Qi Wang[5**], Bostjan Kobe[3], Giuseppe Orsomando[2*‡], Michael P Coleman[1*‡]

[1]John van Geest Centre for Brain Repair, Department of Clinical Neurosciences, University of Cambridge, Cambridge, United Kingdom; [2]Department of Clinical Sciences (DISCO), Section of Biochemistry, Polytechnic University of Marche, Ancona, Italy; [3]School of Chemistry and Molecular Biosciences, Institute for Molecular Bioscience and Australian Infectious Diseases Research Centre, University of Queensland, Brisbane, Australia; [4]Institute for Glycomics, Griffith University, Southport, Australia; [5]Neuroscience, BioPharmaceuticals R and D, AstraZeneca, Waltham, United States

*For correspondence:
al850@cam.ac.uk (AL);
g.orsomando@staff.univpm.it
(GO);
mc469@cam.ac.uk (MPC)

[†] These authors contributed
equally to this work
[‡] These authors contributed
equally to this work

Present address: [§]Ikarovec Ltd,
Norwich Innovation Centre,
Norwich, United Kingdom;
[#]Cambridge Institute for
Medical Research, University
of Cambridge, Cambridge,
United Kingdom; [¶]Vertex
Pharmaceuticals, Boston, United
States; [**]Kymera Therapeutics,
Watertown, United States

Competing interest: See page
20

Reviewing Editor: Moses V
Chao, New York University
Langone Medical Center, New
York City, United States

## Abstract

Axon loss underlies symptom onset and progression in many neurodegenerative disorders. Axon degeneration in injury and disease is promoted by activation of the NAD-consuming enzyme SARM1. Here, we report a novel activator of SARM1, a metabolite of the pesticide and neurotoxin vacor. Removal of SARM1 completely rescues mouse neurons from vacor-induced neuron and axon death in vitro and in vivo. We present the crystal structure of the *Drosophila* SARM1 regulatory domain complexed with this activator, the vacor metabolite VMN, which as the most potent activator yet known is likely to support drug development for human SARM1 and NMNAT2 disorders. This study indicates the mechanism of neurotoxicity and pesticide action by vacor, raises important questions about other pyridines in wider use today, provides important new tools for drug discovery, and demonstrates that removing SARM1 can robustly block programmed axon death induced by toxicity as well as genetic mutation.

## Editor's evaluation

Axon degeneration is activated by injury and through a pathway that involves the SARM1 protein, which possesses NAD[+] cleavage activity. This manuscript definitively identifies the pesticide vacor and its metabolite VMN as an activator of Sarm1. The study works out the mechanism of activation via structural analysis of the allosteric binding site. Because the axon degeneration pathway is activated in a number of neurodegenerative contexts, the insights into the mechanism of action of vacor during neurotoxicity provide avenues for future therapeutic strategies.

## Introduction

Sterile alpha and TIR motif-containing protein 1 (SARM1) plays a central, pro-degenerative role in programmed axon death (including Wallerian degeneration) (*Osterloh et al., 2012*). This axon degeneration pathway is activated in a number of neurodegenerative contexts, including in human disease (*Coleman and Höke, 2020*; *Gilley et al., 2021*; *Huppke et al., 2019*; *Lukacs et al., 2019*). SARM1 has a critical nicotinamide adenine dinucleotide (NAD) cleavage (NADase) activity, which is activated

when its upstream regulator and axon survival factor nicotinamide mononucleotide adenylyltransferase 2 (NMNAT2) is depleted or inactive (*Essuman et al., 2017*; *Gilley et al., 2015*; *Gilley and Coleman, 2010*; *Sasaki et al., 2020a*, *Sasaki et al., 2016*). NMNAT2 loss causes accumulation of its substrate nicotinamide mononucleotide (NMN), which promotes axon degeneration (*Di Stefano et al., 2017*; *Loreto et al., 2020*; *Loreto et al., 2015*). NMN is now known to activate SARM1 NADase activity (*Zhao et al., 2019*). Given that *Sarm1* deletion confers robust axon protection, and even lifelong protection against lethality caused by NMNAT2 deficiency (*Gilley et al., 2017*), SARM1 has become a very attractive therapeutic target to prevent neurodegeneration.

In the past 2 years, there has been intense focus on structure-activity studies of SARM1 to help drive drug development. Recent structural data revealed the existence of an allosteric pocket in SARM1 armadillo-repeat (ARM) domain which is important for the regulation of SARM1 activity. NMN binds to this pocket, leading to SARM1 activation (*Figley et al., 2021*). NAD (*Jiang et al., 2020*) can also bind the same allosteric site, competing with NMN to prevent SARM1 activation. The ratio between NMN:NAD is therefore important to regulate SARM1 activity (*Figley et al., 2021*) and we have recently extended this to show that, at physiological levels, changes in NMN levels have the greater impact on the regulation of SARM1 activity (*Angeletti et al., 2021*). Further understanding how SARM1 activity is regulated by small molecules will help develop ways to block its activation.

Here, we have investigated whether vacor, a disused pesticide and powerful neurotoxin associated with human peripheral and central nervous system disorders (*Gallanosa et al., 1981*; *LeWitt, 1980*) and axon degeneration in rats (*Watson and Griffin, 1987*), causes activation of programmed axon death. Vacor is a pyridine derivative that is metabolised to vacor mononucleotide (VMN) and vacor adenine dinucleotide (VAD) through a two-step conversion by nicotinamide phosphoribosyltransferase (NAMPT) and NMNAT2 (*Figure 1A*). Considering VAD inhibits SARM1 regulator NMNAT2 (*Buonvicino et al., 2018*), we reasoned that vacor may induce SARM1-dependent axon death. We demonstrate that vacor neurotoxicity is entirely mediated by SARM1 such that *Sarm1*⁻ᐟ⁻ neurons and their axons are completely resistant. We unexpectedly find that vacor metabolite VMN directly binds to and activates SARM1, causing neuronal death. This study elucidates the mechanism of action of vacor likely underlying its lethal neurotoxic effect in humans. Vacor use is banned, but structurally related pyridines are widely-used and present in drugs, pesticides and food, raising important questions about their potential toxicity and SARM1 as a mediator of environmental neurotoxicity. Our findings also provide important new tools for drug discovery. To our knowledge, VMN is the most potent activator of SARM1 reported so far, and our structural data reveal essential information on SARM1 regulation which will aid drug development to block SARM1 activation.

## Results

### Vacor causes SARM1-dependent neurite and cell death

Intracellular conversion of vacor into its metabolites has been suggested to contribute to its toxicity (*Buonvicino et al., 2018*). We first confirmed that VMN and VAD are generated in vacor-treated dorsal root ganglion (DRG) mouse neurons (*Figure 1A and B*) and that these and a second neuron type, superior cervical ganglia (SCG) mouse neurons, exhibit rapid, dose-dependent neurite degeneration (*Figure 1C–F*). Consistent with the proposed toxic role for vacor metabolites (*Buonvicino et al., 2018*), both nicotinamide (NAM), which competes with vacor as a preferred substrate for NAMPT, and the NAMPT inhibitor FK866 (*Figure 1A*), delayed vacor-induced neurite degeneration (*Figure 1G–J*). Such competition may explain why NAM was an effective treatment for patients with vacor poisoning (*Gallanosa et al., 1981*).

As hypothesised, vacor failed to induce degeneration of *Sarm1*⁻ᐟ⁻ DRG and SCG neurites (*Figure 2A–D*). This protection was extremely strong, with neurites surviving indefinitely even after multiple vacor doses (*Figure 2—figure supplement 1A,B*). *Sarm1*⁻ᐟ⁻ neurites were not only protected from degeneration, but they also continued to grow normally, even with repeated dosing (*Figure 2E and F*). This mirrors the permanent rescue and continued growth previously reported in *Nmnat2* null axons in the absence of SARM1 (*Gilley et al., 2017*; *Gilley et al., 2015*), suggesting complete efficacy in both toxic and inherited types of neuropathy. It also shows that vacor neurotoxicity is predominantly SARM1-dependent.

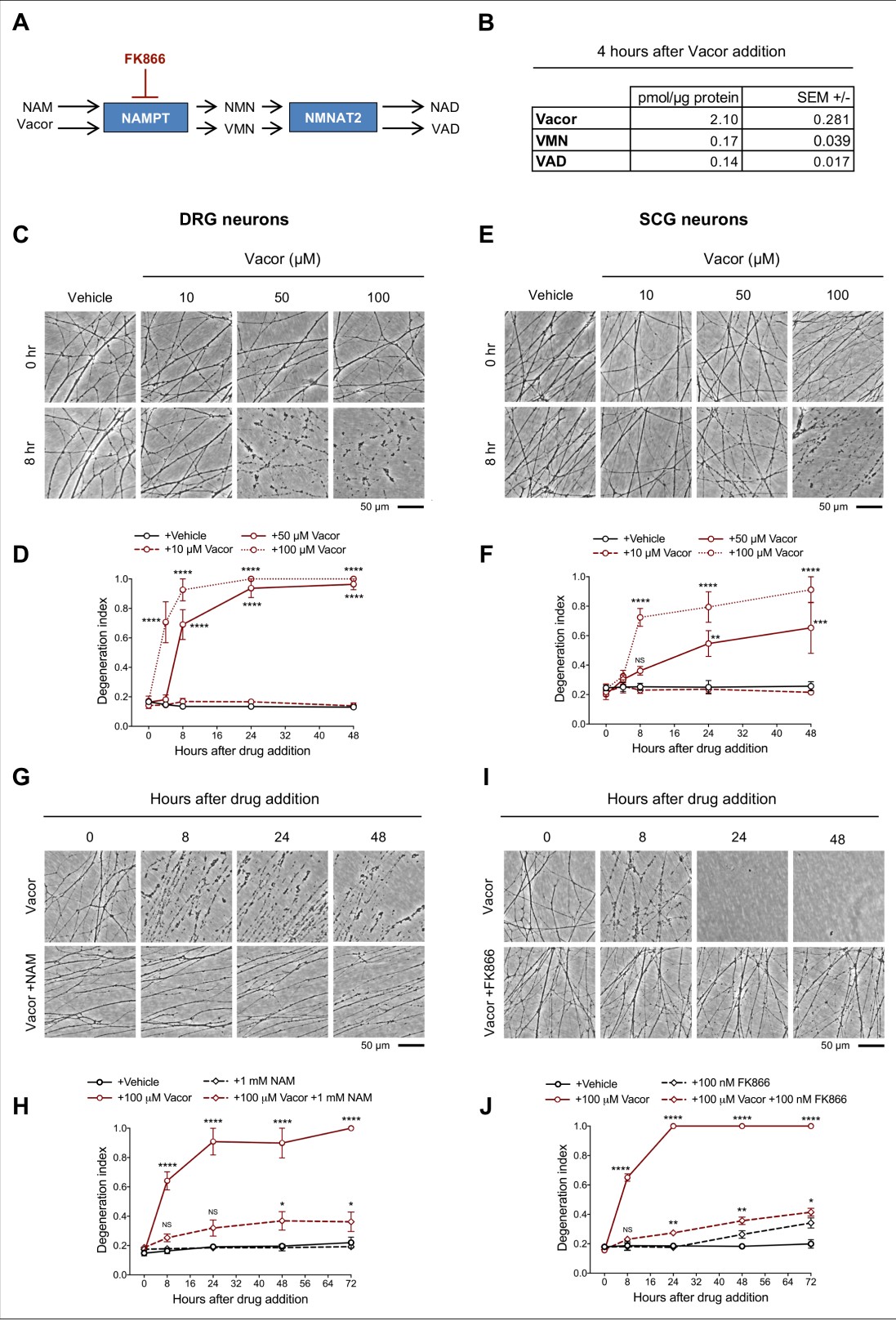

**Figure 1.** Vacor causes neurite degeneration in primary mouse neurons. (**A**) Schematic representation of vacor conversion into VMN and VAD by NAMPT and NMNAT2, respectively. VAD has been reported to inhibit NMNAT2 (*Buonvicino et al., 2018*). Vacor competes with NAM for NAMPT. High doses of NAM or inhibition of NAMPT with FK866 prevent vacor conversion into downstream metabolites (NAM, nicotinamide; NMN, nicotinamide mononucleotide; NAD, nicotinamide adenine dinucleotide; NAMPT, nicotinamide phosphoribosyltransferase; NMNAT2, nicotinamide mononucleotide

*Figure 1 continued on next page*

*Figure 1 continued*

adenylyltransferase 2; VMN, vacor mononucleotide; VAD, vacor adenine dinucleotide). (**B**) Vacor, VMN and VAD levels in wild-type DRG whole explant cultures (neurites and cell bodies) 4 hr after 50 µM vacor treatment (mean ± SEM; n = 3). (**C**) Representative images of neurites from wild-type DRG explant cultures treated with 10, 50, 100 µM vacor or vehicle. (**D**) Quantification of the degeneration index in experiments described in (**C**) (mean ± SEM; n = 3; repeated measures two-way ANOVA followed by Tukey's multiple comparison test; ****, p < 0.0001; statistical significance shown relative to +Vehicle). (**E**) Representative images of neurites from wild-type SCG explant cultures treated with 10, 50, 100 µM vacor or vehicle. (**F**) Quantification of the degeneration index in experiments described in (**E**) (mean ± SEM; n = 3; repeated measures two-way ANOVA followed by Tukey's multiple comparison test; ****, p < 0.0001; ***, p < 0.001; **, p < 0.01; NS, not-significant; statistical significance shown relative to +Vehicle). (**G**) Representative images of neurites from wild-type SCG explant cultures treated with 100 µM vacor or 100 µM vacor +1 mM NAM. (**H**) Quantification of the degeneration index in experiments described in (**G**) (mean ± SEM; n = 3; repeated measures two-way ANOVA followed by Tukey's multiple comparison test; ****, p < 0.0001; *, p < 0.05; NS, not-significant; statistical comparisons shown are: +100 µM Vacor vs +100 µM Vacor +1 mM NAM and +1 mM NAM vs +100 µM Vacor +1 mM NAM). (**I**) Representative images of neurites from wild-type SCG explant cultures treated with 100 µM vacor, 100 nM FK866 or vehicle. (**J**) Quantification of the degeneration index in experiments described in (**I**) (mean ± SEM; n = 3; repeated measures two-way ANOVA followed by Tukey's multiple comparison test; ****, p < 0.0001; **, p < 0.01; *, p < 0.05; NS, not-significant; statistical comparisons shown are: +100 µM Vacor vs +100 µM Vacor +100 nM FK866 and +100 nM FK866 vs +100 µM Vacor +100 nM FK866). Source data for *Figure 1—source data 1*.

The online version of this article includes the following source data for figure 1:

**Source data 1.** Vacor causes neurite degeneration in primary mouse neurons.

SARM1 levels remained relatively stable following vacor treatment (*Figure 2—figure supplement 1C-F*). We therefore tested if SARM1 NADase activity is the critical function required for vacor toxicity. Exogenous expression of wild-type human SARM1 (hSARM1) in *Sarm1⁻/⁻* SCG neurons restored vacor sensitivity, whereas expression of E642A hSARM1, which lacks NADase activity, did not (*Figure 2G–I*; *Figure 2—figure supplement 1G,H*), similar to findings in axotomy (*Essuman et al., 2017*). Interestingly, unlike axotomy, activation of the pathway by vacor also caused SARM1-dependent cell death (*Figure 2I*), similar to that previously reported with constitutively active SARM1 (*Gerdts et al., 2015*; *Gerdts et al., 2013*). To explore this further, we cultured DRG neurons in microfluidic chambers, to allow independent manipulation of the neurite and soma compartments. Addition of vacor to either compartment directly activated local death of cell bodies and/or neurites. However, while vacor-induced cell death also causes neurite degeneration, the reverse was not true: no cell loss was observed after local induction of distal neurite degeneration by vacor (*Figure 2—figure supplement 2A,B*). Intriguingly, the degeneration of neurites in the soma-treated cultures is even more rapid that with direct treatment. Metabolite measurements in neurites after vacor treatment confirmed this involves SARM1 activation (Figure 4D). This suggests there may be components both of conventional Wallerian degeneration following loss of soma support and a possible spreading of SARM1 activation within the cell, possibly as a consequence of the proposed positive feedback upon NAD depletion (*Figley et al., 2021*).

## *Sarm1* deletion confers functional and morphological protection of neurons against vacor toxicity in vivo

We next sought to validate our findings in vivo. To reduce the impact on animal welfare of systemic administration of a rodenticide in mice, we opted for intravitreal (IVT) injection (*Figure 3A*) of vacor. Electroretinogram (ERG) recordings showed that vacor caused a global loss of photoreceptor neuronal activity, and subsequent loss of bipolar and retinal ganglion cell (RGC) responses, which were fully rescued by SARM1 deficiency (*Figure 3B and C*). Vacor administration also caused inner retina neuronal death, as revealed by a significant reduction in RGC number. Again, this was completely prevented by SARM1 deficiency (*Figure 3D and E*). DMSO had no adverse effect on retinal cell survival or function compared to PBS-injected eyes (*Figure 3—figure supplement 1A,B*). These data demonstrate that the absence of SARM1 confers functional and morphological protection of neurons against vacor toxicity in vivo.

## Vacor treatment leads to SARM1 activation

Activation of SARM1 depletes NAD and generates cADPR in neurons (*Essuman et al., 2017*; *Gerdts et al., 2015*; *Sasaki et al., 2020a*, *Sasaki et al., 2016*). Based on the previously reported inhibition of NMNAT2 by VAD (*Buonvicino et al., 2018*), we initially hypothesised that this causes a rise in NMN, the physiological NMNAT2 substrate and a known activator of SARM1 (*Di Stefano et al., 2017*; *Di*

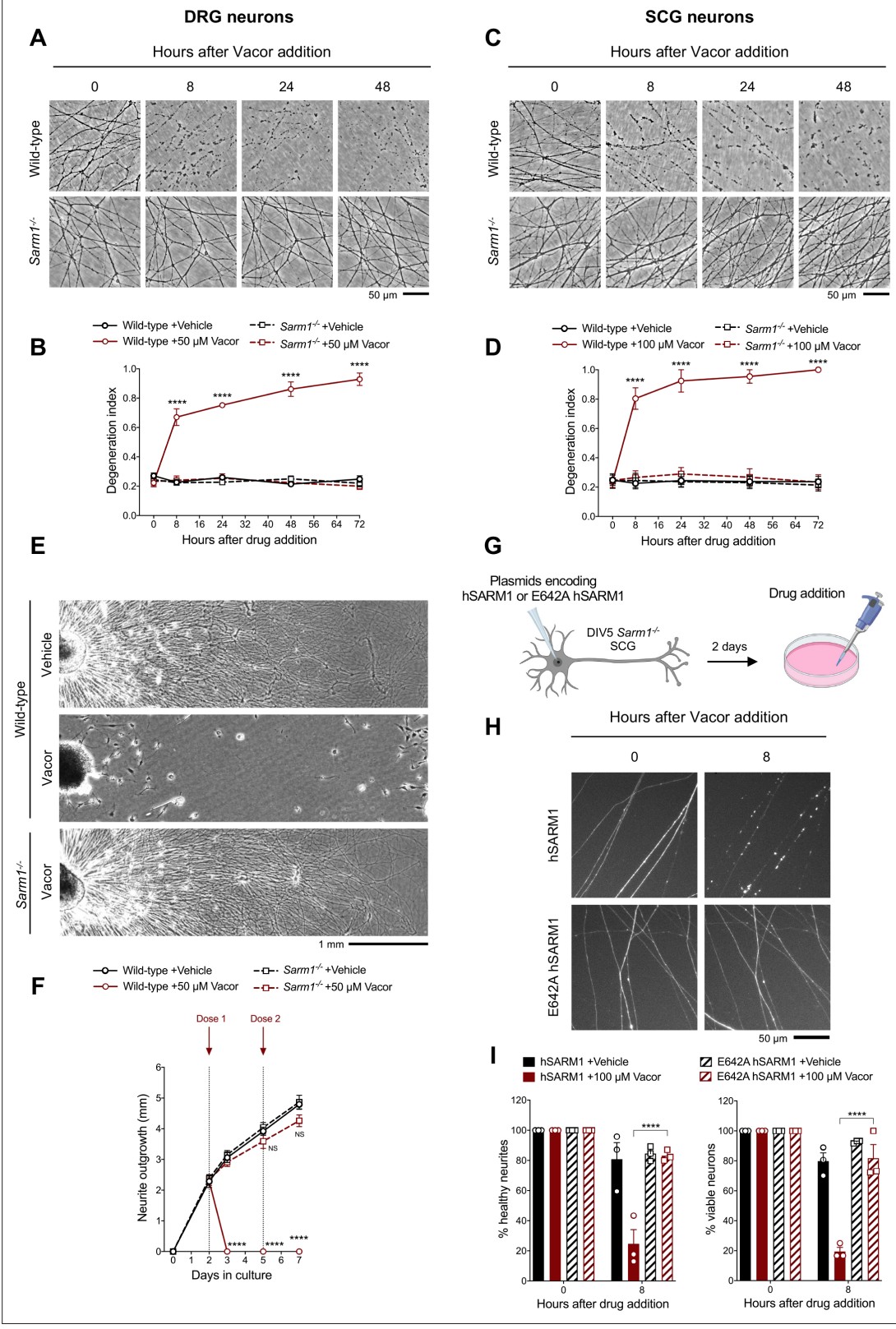

**Figure 2.** Vacor causes SARM1-dependent neurite and cell death. (**A**) Representative images of neurites from wild-type and *Sarm1*⁻/⁻ DRG (littermates) explant cultures treated with 50 μM vacor. (**B**) Quantification of the degeneration index in experiments described in (**A**) (mean ± SEM; n = 4; repeated measures three-way ANOVA followed by Tukey's multiple comparison test; ****, p < 0.0001; statistical significance shown relative to *Sarm1*⁻/⁻ +50 μM Vacor). (**C**) Representative images of neurites from wild-type and *Sarm1*⁻/⁻ SCG explant cultures treated with 100 μM vacor. (**D**) Quantification of the

*Figure 2 continued on next page*

*Figure 2 continued*

degeneration index in experiments described in (**C**) (mean ± SEM; n = 4; repeated measures three-way ANOVA followed by Tukey's multiple comparison test; ****, p < 0.0001; statistical significance shown relative to *Sarm1*⁻/⁻ +100 μM Vacor). (**E**) Representative images of neurite outgrowth at DIV7 from wild-type and *Sarm1*⁻/⁻ DRG explant cultures treated with 50 μM vacor or vehicle. Multiple doses of vacor or vehicle were added at DIV2 and DIV5. (**F**) Quantification of neurite outgrowth in (**E**) (mean ± SEM; n = 5; repeated measures three-way ANOVA followed by Tukey's multiple comparison test; ****, p < 0.0001; NS, not-significant; statistical significance shown relative to Wild-type +Vehicle). (**G**) Schematic representation of the experimental design for (**H**) ('Created with BioRender'). (**H**) Representative images of neurites from *Sarm1*⁻/⁻ SCG dissociated neurons co-injected with plasmids encoding wild-type or E642A hSARM1 and DsRed (to label neurites) and treated with 100 μM vacor. (**I**) Quantification of healthy neurites and viable neurons in experiments in (**H**) is shown as a percentage relative to 0 hr (time of drug addition) (mean ± SEM; n = 3; repeated measures three-way ANOVA followed by Tukey's multiple comparison test; ****, p < 0.0001). Source data for *Figure 2—source data 1*.

The online version of this article includes the following source data and figure supplement(s) for figure 2:

**Source data 1.** Vacor causes SARM1-dependent neurite and cell death.

**Figure supplement 1.** Long-term survival of *Sarm1*⁻/⁻ SCG neurites following multiple vacor doses.

**Figure supplement 1—source data 1.** Long-term survival of Sarm1⁻/⁻ SCG neurites following multiple vacor doses.

**Figure supplement 2.** Local death of neurites and cell bodies caused by vacor.

**Figure supplement 2—source data 1.** Local death of neurites and cell bodies caused by vacor.

*Stefano et al., 2015*; *Loreto et al., 2015*; *Zhao et al., 2019*), thereby stimulating SARM1 activity to trigger vacor-dependent axon death. Surprisingly, while SARM1 activation was confirmed by a drastic SARM1-dependent decline in NAD and increase in cADPR following vacor administration, NMN did not rise and even fell slightly (*Figure 4A–D*). Furthermore, NMN and NAD levels and their ratio to one another were not altered in our vacor-treated *Sarm1*⁻/⁻ neurons (*Figure 4A and B*), suggesting that the reported inhibition of NAMPT and NMNAT2 by vacor/VAD (*Buonvicino et al., 2018*) does not occur in this context at this drug concentration. Crucially, NMN deamidase, an enzyme that strongly preserves axotomised axons by preventing NMN accumulation (*Figure 4E*; *Figure 4—figure supplement 1A*, *Di Stefano et al., 2017*; *Di Stefano et al., 2015*; *Loreto et al., 2015*), was also unable to protect against vacor toxicity (*Figure 4E and F*), even though recombinant NMN deamidase retains its ability to convert NMN to nicotinic acid mononucleotide (NaMN) in the presence of vacor, VMN and VAD (*Figure 4—figure supplement 1B*). WLD^S, another enzyme that limits NMN accumulation (*Di Stefano et al., 2015*), also failed to rescue neurons against vacor neurotoxicity (*Figure 4—figure supplement 1C,D*). These data suggest that, in this specific context at least, NMN is not responsible for SARM1 activation.

## Vacor metabolite VMN potently activates SARM1

What did accumulate in vacor-treated neurons was the vacor metabolite VMN (*Figure 1B*) and, given its structural similarity to the endogenous SARM1 activator NMN (*Figure 5—figure supplement 1A*), we hypothesised that VMN might instead directly activate SARM1, leading to cell and axon death. Crucially, we found that VMN potently activates the NADase activity of recombinant hSARM1, even more so than NMN, having a lower $K_a$ and resulting in a greater induction (*Figure 5A*; *Figure 5—figure supplement 1B*). Conversely, VAD only had a weak inhibitory effect on hSARM1 activity (*Figure 5—figure supplement 1C*). Intriguingly, recombinant hSARM1 NADase activity dropped at higher VMN concentrations (*Figure 5A*). Notwithstanding the doses of vacor used in this study resulting in VMN levels in neurons within the activation range of SARM1, this inhibitory action could reveal critical information on how SARM1 activity is regulated. Analysis via best fitting (see equation in Materials and methods) suggests that our data are compatible with two distinct binding sites of VMN on SARM1 and led us to establish binding constants for activation ($K_a$) and inhibition ($K_i$) (*Figure 5A*).

While NMN activates SARM1 (*Zhao et al., 2019*) and promotes programmed axon death when NMNAT2 is depleted (*Di Stefano et al., 2017*; *Loreto et al., 2020*; *Loreto et al., 2015*), exogenous NMN does not induce degeneration of uninjured neurites (*Di Stefano et al., 2015*), probably because it is rapidly converted to NAD when NMNAT2 is present. However, we now show that exogenous application of its analogue, VMN, does induce SARM1-dependent death of uninjured DRG neurites (*Figure 5B and C*). This difference likely reflects a combination of VMN being both a more potent activator of SARM1 and it accumulating more because it is not efficiently metabolised by NMNAT2 (*Buonvicino et al., 2018*). VMN-dependent activation of SARM1 being the effector of vacor toxicity

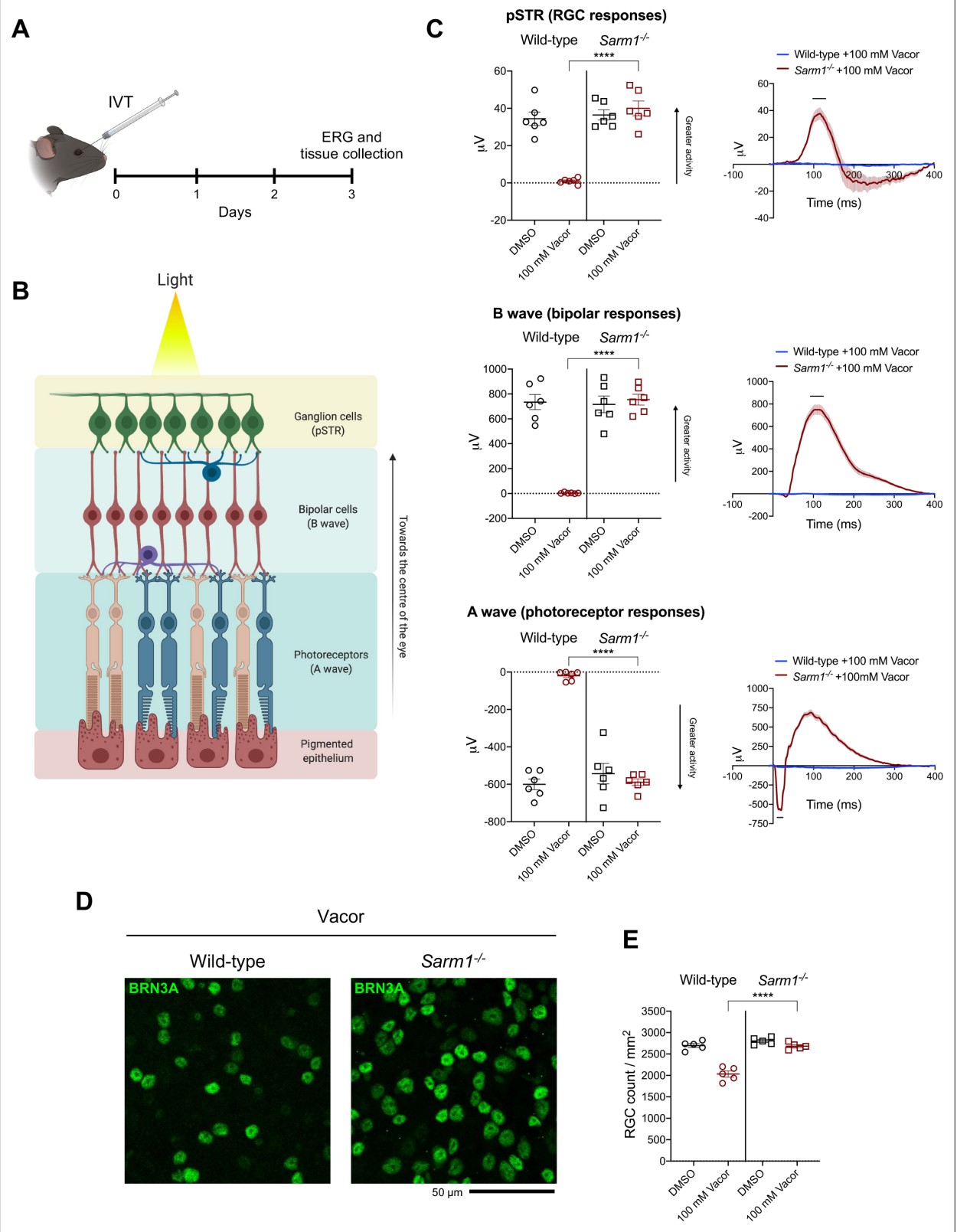

**Figure 3.** *Sarm1* deletion confers functional and morphological protection of neurons against vacor toxicity in vivo. (**A**) Schematic representation of the experimental design for (**C,D**) ('Created with BioRender'). (**B**) Graphic illustration of the different retinal layers ('Created with BioRender'). (**C**) Quantification of the ERG responses (pSTR, B wave and A wave) from wild-type and *Sarm1⁻/⁻* mice injected with 100 mM vacor or DMSO (vehicle) (mean ± SEM; n = 6; two-way ANOVA followed by Tukey's multiple comparison test; ****, p < 0.0001). (**D**) Representative images of RGC from wild-type and

*Figure 3 continued on next page*

*Figure 3 continued*

*Sarm1^{-/-}* mice injected with 100 mM vacor or DMSO (vehicle). (**E**) Quantification of RGC numbers from wild-type and *Sarm1^{-/-}* mice injected with 100 mM vacor or DMSO (vehicle) (mean ± SEM; n = 5; repeated measures two-way ANOVA followed by Tukey's multiple comparison test; ****, p < 0.0001). Source data for *Figure 3—source data 1*.

The online version of this article includes the following source data and figure supplement(s) for figure 3:

**Source data 1.** Sarm1 deletion confers functiona and morphological protection of neurons against vacor toxicity in vivo.

**Figure supplement 1.** DMSO has no adverse effect on retinal cell survival or function compared to PBS-injected eyes.

**Figure supplement 1—source data 1.** DMSO has no adverse effect on retinal cell survival or function compared to PBS-injected eyes.

is also supported by several other findings. First, unlike NMN, VMN is not a substrate of NMN deamidase (*Figure 5—figure supplement 1D*), thus explaining its inability to protect against vacor neurotoxicity (*Figure 4E and F*). In addition, VMN is not a substrate of the nuclear isoform NMNAT1 and is a poor substrate of mitochondrial NMNAT3 (*Buonvicino et al., 2018*), so VMN accumulation in cell bodies (*Figure 5—figure supplement 1E*) and subsequent SARM1 activation, as detected by a rise of cADPR in this compartment (*Figure 4D*), provides a clear rationale for why vacor administration also causes cell death.

## VMN activates SARM1 through direct binding to SARM1 ARM domain

SARM1 ARM domain is believed to maintain SARM1 in an inhibited state (*Figley et al., 2021*; *Gerdts et al., 2013*; *Jiang et al., 2020*). SARM1 is activated by NMN binding directly to the ARM domain (*Figley et al., 2021*; *Gu et al., 2021*). We therefore hypothesised that VMN could directly interact with the ARM domain to release this inhibition and activate SARM1, causing neuronal death. Because we were unable to produce hSARM1^ARM recombinantly, we used the ARM domain of *Drosophila* SARM1 (dSARM1^ARM, residues 315–678), produced in *E. coli*, to investigate if VMN interacts directly with dSARM1^ARM. Isothermal titration calorimetry (ITC) analysis showed that VMN bound to dSARM1^ARM with a $K_d$ value of 2.83 ± 0.16 µM (1:1 molar ratio) (*Figure 6—figure supplement 1A*). To further characterise this interaction, we determined the crystal structure of the VMN-bound dSARM1^ARM (1.69 Å resolution). dSARM1^ARM contains eight tandem armadillo repeats (ARM1-8) stacking into an unusually compact right-handed superhelix (*Figure 6A*; *Supplementary file 1*). The bound VMN molecule sits in the central groove of dSARM1^ARM, which is mainly lined by the H3 helices from ARM1-5 (*Figure 6A*; *Figure 6—figure supplement 1B*). The interaction is mediated by two pi-stacking interactions between the indole of W385 and the pyridine moiety of VMN, and between the imidazole ring of H392 and the nitrobenzene moiety of VMN (*Figure 6B*; *Figure 6—figure supplement 2A,B*); and eight hydrogen bonds, with a buried surface area of ~910 Å² (*Figure 6B*; *Figure 6—figure supplement 2A,B*). The hydroxyl group from the side chain of Y396 and the amino groups from the side chains of the conserved R437 and K476 form hydrogen bonds with the phosphate group of VMN; the amino group from the side chain of N640 forms a hydrogen bond with the nitro group of VMN; and the imidazole rings of H392 and H473 form hydrogen bonds with the urea and ribose of VMN, respectively (*Figure 6B*; *Figure 6—figure supplement 2A,B*).

Our data show that VMN has over two-fold higher affinity for dSARM1^ARM than NMN ($K_d$ = 6.39 ± 0.04 µM), whose own interaction with the ARM domain was recently reported (*Figure 6—figure supplement 1A*, *Figley et al., 2021*). VMN-bound dSARM1^ARM adopts a similar conformation to NMN-bound dSARM1^ARM (*Figure 6C*). Comparison of the two structures reveals that the NMN moiety of VMN and NMN share the same binding mode, with the key residues involved in NMN binding interacting with the analogous portions of the VMN molecule (*Figure 6B*). Importantly, in the NMN-bound structure, NMN forms a hydrogen bond with the residues H599 and G600 (loop connecting ARM6 and 7), while in the VMN-bound structure, due to the extra portion of VMN (urea and nitrobenzene groups) extending to the region between the N- and C-termini of the protein, two different hydrogen bonds are formed between VMN and H392 (ARM2 H1) at the N-terminus, and N640 (ARM8 H1) at the C-terminus. Furthermore, the pi-stacking interaction between H392 and the nitrobenzene of VMN is specific to this compound. These different and extra interactions VMN forms could explain why VMN induces SARM1 activation more efficiently, compared to NMN. The compaction seen in both NMN and VMN-bound dSARM1^ARM structures likely represents the conformation of the ARM domain in active SARM1, based on the comparison with the more open crystal structures of ligand-free and

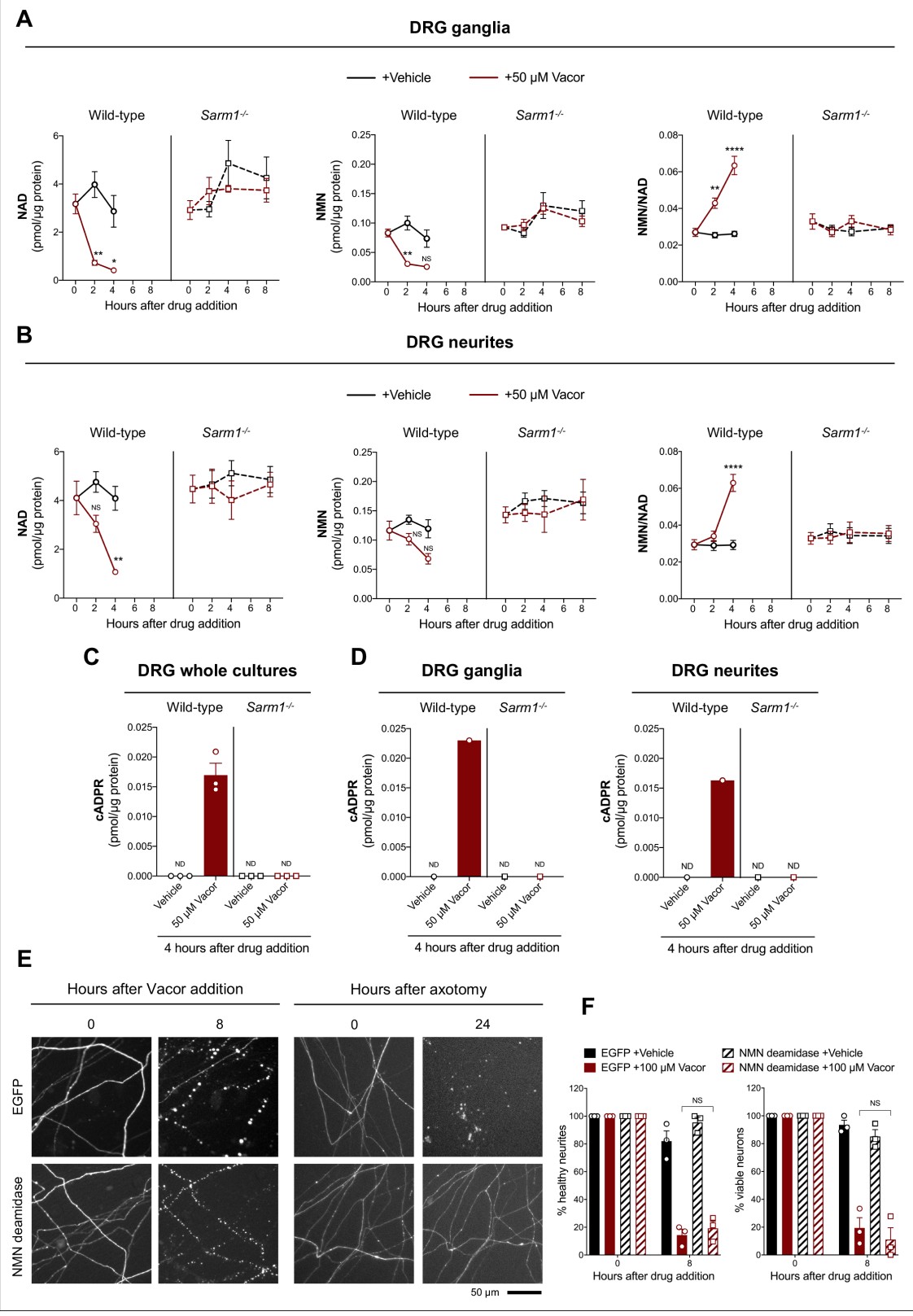

**Figure 4.** Vacor treatment leads to SARM1 activation. (**A,B**) NMN and NAD levels and NMN/NAD ratio in ganglia (**A**) and neurite (**B**) fractions from wild-type and *Sarm1*[-/-] DRG explant cultures at the indicated time points after 50 µM vacor or vehicle treatment (mean ± SEM; n = 4; three-way ANOVA followed by Tukey's multiple comparison test; ****, $p < 0.0001$; **, $p < 0.01$; *, $p < 0.05$; NS, not-significant). (**C**) cADPR levels in wild-type and *Sarm1*[-/-] DRG whole explant cultures (neurites and cell bodies) 4 hr after 50 µM vacor or vehicle treatment. cADPR levels were consistently above the detection

*Figure 4 continued on next page*

*Figure 4 continued*

limit (~1 fmol/µg protein) only in wild-type DRG explant cultures treated with vacor (mean ± SEM; n = 3; ND, not-detectable). (**D**) A single analysis of cADPR levels in ganglia and neurite fractions from wild-type and *Sarm1⁻/⁻* DRG explant cultures 4 hr after 50 µM vacor or vehicle treatment (n = 1). (**E**) Representative images of neurites from *Sarm1⁻/⁻* SCG dissociated neurons co-injected with plasmids encoding hSARM1, EGFP or EGFP-NMN deamidase and DsRed (to label neurites) and treated with 100 µM vacor. As an experimental control, *Sarm1⁻/⁻* SCG dissociated neurons injected with the same injection mixtures were axotomised. As expected, neurites expressing NMN deamidase were still intact 24 hr after axotomy. (**F**) Quantification of healthy neurites and viable neurons in experiments in (**E**) is shown as a percentage relative to 0 hr (time of drug addition) (mean ± SEM; n = 3; repeated measures three-way ANOVA followed by Tukey's multiple comparison test; NS, not-significant). Source data for ***Figure 4—source data 1***.

The online version of this article includes the following source data and figure supplement(s) for figure 4:

**Source data 1.** Vacor treatment leads to SARM1 activation.

**Figure supplement 1.** Effect of vacor, VMN and VAD on recombinant NMN deamidase activity and lack of protection after vacor treatment in *Wldˢ* neurons.

**Figure supplement 1—source data 1.** Effect of vacor, VMN and VAD on recombinant NMN deamidase activity and lack of protection after vacor treatment in WldS neurons.

---

NaMN-bound dSARM1ᴬᴿᴹ, as well as cryo-EM (electron microscopy) structures of ligand-free and NAD-bound hSARM1, representing the inactive state of the protein (***Bratkowski et al., 2020***; ***Figley et al., 2021***; ***Sporny et al., 2020***) (***Figure 6C***). NaMN stabilises the open conformation of dSARM1ᴬᴿᴹ due to the rotation of its nicotinic acid (NA) portion towards the N-terminus of the protein, and is therefore unable to form any interactions with the C-terminal region of the protein (***Sasaki et al., 2021***). Although the corresponding pyridine moiety of VMN also has a similar rotation, the extended portion of VMN permits the interaction with the C-terminal region of the protein, hence maintaining a compact conformation (***Figure 6C***). In the NAD-bound hSARM1ᴬᴿᴹ structure, the portion of NAD equivalent to NMN and VMN interacts with the protein through similar interactions, but the adenine group and the adjacent ribose of NAD extend further down to the side of the region between the termini. This would cause steric clashes with the C-terminal region of the protein, particularly with the loop connecting ARM6 and 7 (residues 594–602), thus preventing NAD from inducing a more compact conformation in the ARM domain (***Figure 6—figure supplement 1C***), keeping the protein in the inactive state. In summary, our data suggest that VMN activates SARM1 with a mechanism similar to NMN, but with some important differences, inducing a compaction of the ARM domain, destabilising its interfaces with the neighbouring ARM, SAM, and TIR domains, and eventually permitting the TIR domains to self-associate and hydrolyse NAD.

## Mutations in the VMN binding pocket of hSARM1 ARM domain prevent vacor toxicity

To validate the interactions observed in the crystal structure, we performed mutagenesis studies, focusing on the interactions associated with W385 (human W103), R437 (human R157), and K476 (human K193), because these residues are highly conserved and specifically interact with VMN through their side chains. We introduced the corresponding W103A, R157A, and K193R mutations into human SARM1 and exogenously expressed them in *Sarm1⁻/⁻* SCG mouse neurons (***Figure 7—figure supplement 1A***). Unlike wild-type hSARM1, these mutants failed to restore neurite degeneration and cell death after vacor administration (***Figure 7A–F***). We picked K193R hSARM1 for further biochemical characterisation, as it is known to cause loss-of-function, based on a previous study (***Geisler et al., 2019***). Crucially, neither VMN nor NMN could activate the NADase activity of recombinant K193R hSARM1 (***Figure 7G***).

## Discussion

Overall, our study provides clear evidence that direct SARM1 activation by the vacor metabolite VMN underlies vacor neurotoxicity. Crucially, the comparison between VMN and NMN-bound dSARM1ᴬᴿᴹ structures and the differences from the ligand-free and NAD-bound hSARM1ᴬᴿᴹ structures provide the explanation for the mechanism of how these ligands target the same allosteric site, but regulate SARM1 activity differently. The data not only provide support for a key physiological role of NMN in the regulation of SARM1 activity and axon degeneration, but given VMN's greater potency than NMN, we anticipate that VMN and vacor will be important tools to support drug discovery in several

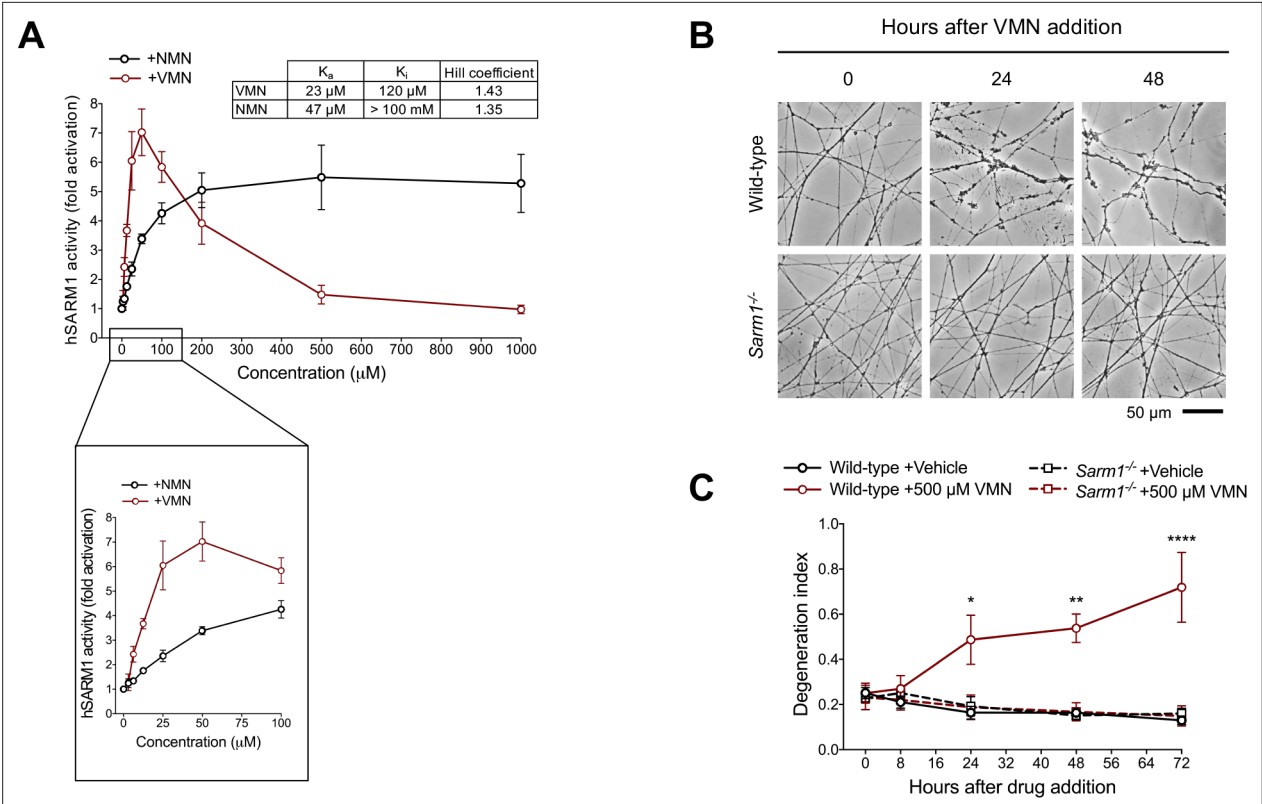

**Figure 5.** Vacor metabolite VMN potently activates SARM1. (**A**) Fold change of NADase activity of purified, recombinant hSARM1 in the presence of NMN and VMN (mean ± SEM; n = 3). hSARM1 average basal activity is 18.12 ± 3.02 milliU/mg (fold activation = 1). Rates are relative to controls measured with 250 µM NAD alone. Both NMN and VMN, once added to each reaction mixture, were not consumed during incubation. Experimental data were fitted to the modified Michaelis-Menten equation in Methods to calculate the kinetic parameters shown in the attached table. Hill coefficients for NMN and VMN indicate positive cooperativity in binding in both cases. Best fitting also revealed a $K_m$ for NAD of 70 µM. (**B**) Representative images of neurites from wild-type and *Sarm1⁻/⁻* DRG (littermates) explant cultures treated with 500 µM VMN. (**C**) Quantification of the degeneration index in experiments described in (**B**) (mean ± SEM; n = 3; repeated measures three-way ANOVA followed by Tukey's multiple comparison test; ****, $p < 0.0001$; **, $p < 0.01$; *, $p < 0.05$; statistical significance shown relative to *Sarm1⁻/⁻* +500 µM VMN). Source data for ***Figure 5—source data 1***.

The online version of this article includes the following source data and figure supplement(s) for figure 5:

**Source data 1.** Vacor metabolite VMN potently activates SARM1.

**Figure supplement 1.** Representative basal NADase rate of recombinant hSARM1.

**Figure supplement 1—source data 1.** Representative basal NADase rate of recombinant hSARM1.

**Figure supplement 2.** VMN and VAD synthesis and purification.

ways. The identification of key residues in the ARM domain involved in the potent activation of SARM1 by VMN should facilitate rational drug design; further understanding why VMN activates SARM1 more potently than NMN could lead to the development of ligands with higher-affinity and selectivity to lock SARM1 in its inactive state. It will be important to expand also on VMN inhibition functions. Finally, we have also recently reported that recombinant hSARM1 uses vacor for base exchange to produce VAD (***Angeletti et al., 2021***), suggesting that vacor can also directly interact with SARM1. Understanding whether base exchange is important for SARM1 pro-degenerative function is another important area of research.

Vacor is currently the most effective chemical to directly activate SARM1 and cause neuronal death, eliminating the need for complex axotomy experiments or the use of drugs with non-specific actions. The IVT vacor model we have developed could also be used for relatively rapid validation of drugs targeting SARM1 in vivo in rodents, with both morphological and functional readouts. Interestingly, it was previously reported that *Sarm1* deletion does not prevent RGC death following optic nerve crush in mice (***Fernandes et al., 2018***). However, our data show that SARM1 activation is sufficient to drive degeneration of multiple types of retinal neuron (including RGC) in addition to photoreceptors

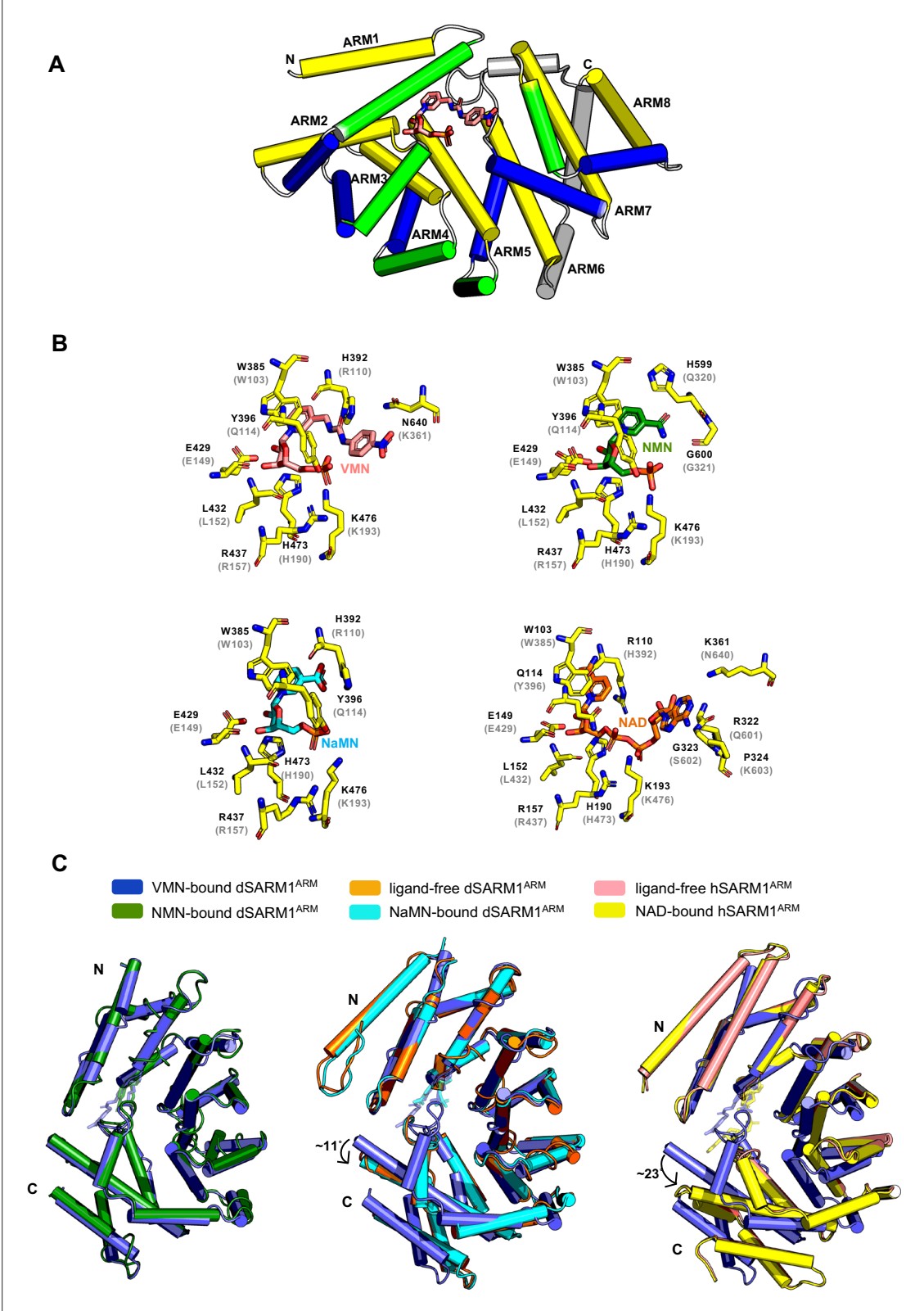

**Figure 6.** VMN activates SARM1 through direct binding to SARM1 ARM domain. (**A**) Crystal structure of VMN-bound dSARM1^ARM. dSARM1^ARM contains eight armadillo motifs (ARM1-8). Except for ARM1 (containing only the H3 helix) and ARM7 (containing only H2 and H3 helices), other motifs consist of H1, H2, and H3 helices, coloured green, blue, and yellow, respectively. The unusual ARM6, which makes a sharp turn at its C-terminus, is coloured grey. Nitrogen, oxygen and phosphorous are coloured blue, red, and orange, respectively, in the VMN molecule. (**B**) Stick representation of the interaction

*Figure 6 continued on next page*

*Figure 6 continued*

of dSARM1$^{ARM}$ with VMN (pink), NMN (PDB: 7LCZ; green) and NaMN (PDB: 7RTC; cyan), and hSARM1 ARM domain with NAD (PDB: 7CM6; orange). The corresponding residues in *Drosophila* or human SARM1$^{ARM}$ are shown in parentheses. Nitrogen, oxygen, and phosphorous are coloured blue, red, and orange. (**C**) Structural comparison of the ARM domains bound to different ligands. The panel on the left shows the structural superposition of the N-terminal regions (residues 373–444) of VMN and NMN-bound dSARM1$^{ARM}$ (PDB: 7LCZ; RMSD is 0.4 Å over 304 Cα atoms). The middle panel shows the superposition of the N-terminal regions (residues 373–444) of VMN-bound dSARM1$^{ARM}$, ligand-free (PDB: 7LCY) and NaMN-bound dSARM1$^{ARM}$ (PDB: 7RTC, RMSD between ligand-free and VMN-bound structures is 1.6 Å over 300 Cα atoms; RMSD between NaMN-bound and VMN-bound structures is 1.7 Å over 300 Cα atoms). Panel on the right shows the superposition of the N-terminal regions (residues 373–444 of dSARM1$^{ARM}$ and residues 91–164 of hSARM1$^{ARM}$) of VMN-bound dSARM1$^{ARM}$, unbound (PDB: 7CM5) and NAD-bound hSARM1$^{ARM}$ (PDB: 7CM6, RMSD between ligand-free and VMN-bound structures is 4.3 Å over 300 Cα atoms; RMSD between NAD-bound and VMN-bound structures is 4.3 Å over 300 Cα atoms).

The online version of this article includes the following figure supplement(s) for figure 6:

**Figure supplement 1.** Analysis of dSARM1$^{ARM}$: VMN interaction.

**Figure supplement 2.** Sequence alignment of SARM1 orthologs.

(*Sasaki et al., 2020b*), making SARM1 a promising target to treat retinal degeneration, at least in response to certain insults. To explain these differences, we suggest that retrograde RGC soma death after axotomy proceeds by a different, SARM1-independent mechanism such as loss of retrogradely delivered trophic factors.

Our data also further implicate programmed axon death in human disease. It appears that this degeneration pathway can be aberrantly activated in humans not only by genetic mutation of *NMNAT2* (*Huppke et al., 2019*; *Lukacs et al., 2019*), but also by SARM1 activation in severe neurotoxicity within hours of vacor ingestion (*LeWitt, 1980*). Although vacor use is banned, many other pyridines are widely used in our environment, and even directly administered to humans or ingested as pharmaceuticals, perfumes or food additives (*Adams and De Kimpe, 2006*; *Li et al., 2016*), raising important questions about their potential to activate SARM1 and cause axon degeneration or neuronal death. Indeed, one other nicotinamide analog, 3-acetyl pyridine, is already associated with axon degeneration (*Desclin and Escubi, 1974*) and subsequent to our preprint of this work has now been reported to act through SARM1 in a similar manner (*Wu et al., 2021*). Genetic hyperactivation of SARM1 has also now been associated with amyotrophic lateral sclerosis (*Bloom et al., 2021*; *Gilley et al., 2021*).

Despite the extreme toxicity of vacor, SARM1 deficiency completely rescues neurons and preserves their functionality. Together with the previously reported lifelong protection in a genetic model of a rare human axonopathy (*Gilley et al., 2017*; *Lukacs et al., 2019*), this further supports the therapeutic potential of blocking SARM1 to completely prevent programmed axon death. Lastly, as vacor causes pancreatic β-cell destruction and diabetes in humans (*Gallanosa et al., 1981*), these findings could also have broader implications, uncovering a role for SARM1 in the survival of insulin producing cells and aiding research on diabetes. In conclusion, our study identifies a powerful SARM1 activator, advancing our understanding of how SARM1 activity is regulated and providing key support to drug discovery targeting programmed axon death.

## Materials and methods

All studies conformed to the institution's ethical requirements in accordance with the 1986 Animals (Scientific Procedures) Act under Project Licences PPL P98A03BF9 and PP1824519, and in accordance with the Association for Research in Vision and Ophthalmology's Statement for the Use of Animals in Ophthalmic and Visual Research.

### Primary neuronal cultures

DRG ganglia were dissected from C57BL/6J (RRID:IMSR_JAX:000664), *Sarm1*$^{-/-}$ (RRID:MGI:3765957) and *Wld*$^{S}$ (RRID:MGI:4438866) E13.5-E14.5 mouse embryos and SCG ganglia were dissected from postnatal day 0–2 mouse pups. Littermates from *Sarm1*$^{+/-}$ crosses were used when possible, as indicated in figure legends. Explants were cultured in 35 mm tissue culture dishes pre-coated with poly-L-lysine (20 µg/ml for 1 hr; Merck) and laminin (20 µg/ml for 1 hr; Merck) in Dulbecco's Modified Eagle's Medium (DMEM, Gibco) with 1 % penicillin/streptomycin, 50 ng/ml 2.5 S NGF (all Invitrogen) and 2 % B27 (Gibco). Four µM aphidicolin (Merck) was used to reduce proliferation and viability of small numbers of non-neuronal cells (*Loreto and Gilley, 2020*). For cultures of dissociated SCG neurons,

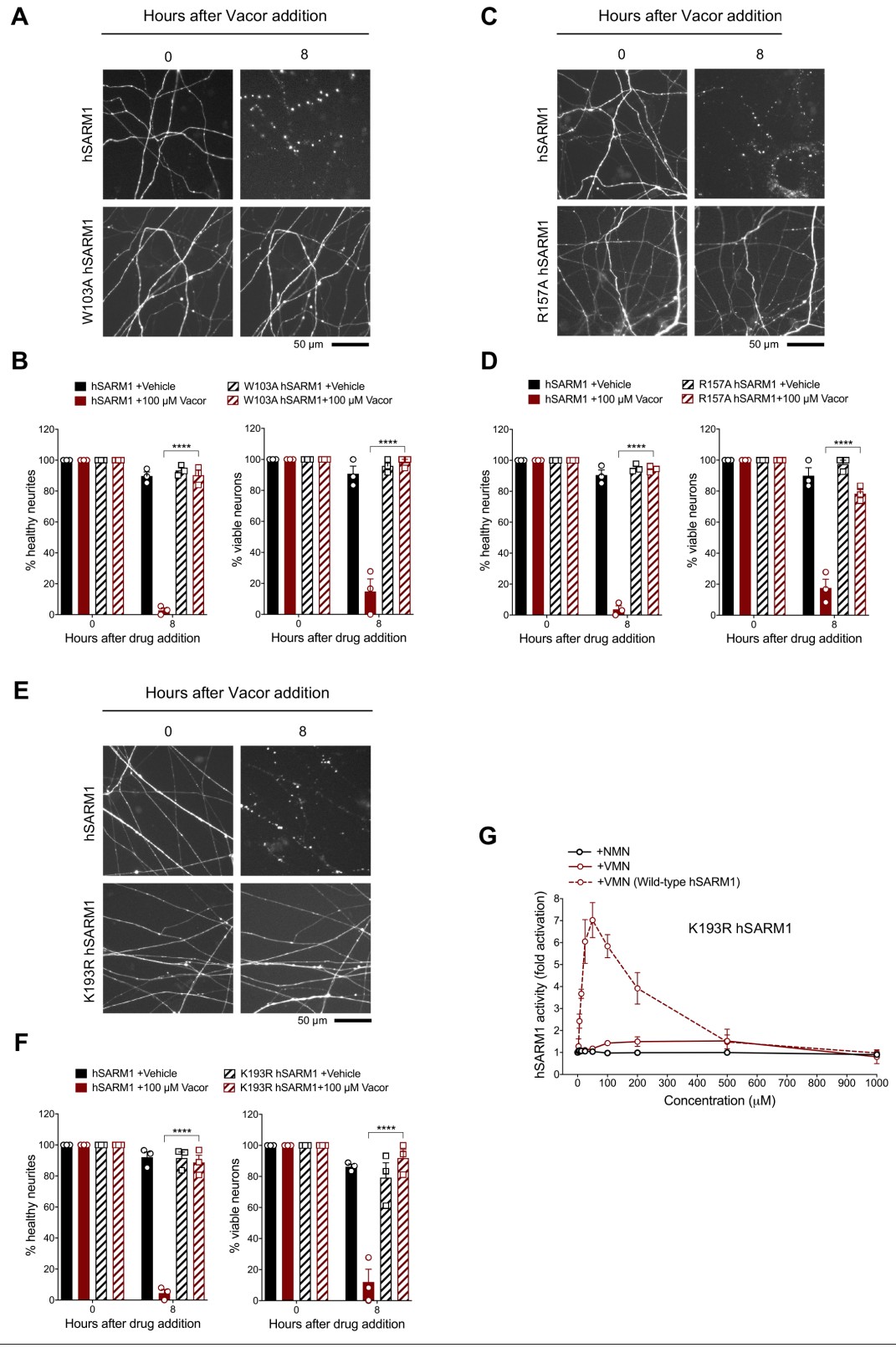

**Figure 7.** Mutations in the VMN binding pocket of hSARM1 ARM domain prevent vacor toxicity. (**A**) Representative images of neurites from *Sarm1⁻/⁻* SCG dissociated neurons co-injected with plasmids encoding wild-type or W103A hSARM1 and DsRed (to label neurites) and treated with 100 μM vacor. (**B**) Quantification of healthy neurites and viable neurons in experiments in (**A**) is shown as a percentage relative to 0 hr (time of drug addition) (mean ± SEM;

*Figure 7 continued on next page*

*Figure 7 continued*

n = 3; repeated measures three-way ANOVA followed by Tukey's multiple comparison test; ****, p < 0.0001. (**C**) Representative images of neurites from *Sarm1*-/- SCG dissociated neurons co-injected with plasmids encoding wild-type or R157A hSARM1 and DsRed (to label neurites) and treated with 100 µM vacor. (**D**) Quantification of healthy neurites and viable neurons in experiments in (**C**) is shown as a percentage relative to 0 hr (time of drug addition) (mean ± SEM; n = 3; repeated measures three-way ANOVA followed by Tukey's multiple comparison test; ****, p < 0.0001). (**E**) Representative images of neurites from *Sarm1*-/- SCG dissociated neurons co-injected with plasmids encoding wild-type or K193R hSARM1 and DsRed (to label neurites) and treated with 100 µM vacor. (**F**) Quantification of healthy neurites and viable neurons in experiments in (**E**) is shown as a percentage relative to 0 hr (time of drug addition) (mean ± SEM; n = 3; repeated measures three-way ANOVA followed by Tukey's multiple comparison test; ****, p < 0.0001). (**G**) Fold change of NADase activity of purified, recombinant K193R hSARM1 in the presence of NMN and VMN (wild-type hSARM1+ VMN is also shown for comparison) (mean ± SEM; n = 2–3). K193R hSARM1 average basal activity is 17.75 ± 2.47 milliU/mg (fold activation = 1). Source data for *Figure 7— source data 1*.

The online version of this article includes the following source data and figure supplement(s) for figure 7:

**Source data 1.** Mutations in the VMN binding pocket of hSARM1 ARM domain prevent vacor toxicity.

**Figure supplement 1.** Expression of wild-type, W103A, R157A and K193R hSARM1 in SCG neurons.

*Sarm1*-/- SCG ganglia were incubated in 0.025 % trypsin (Merck) in PBS (without $CaCl_2$ and $MgCl_2$) (Merck) for 30 min, followed by incubation with 0.2 % collagenase type II (Gibco) in PBS for 20 min. Ganglia were then gently dissociated using a pipette. Dissociated neurons were plated in a poly-L-lysine and laminin-coated area of ibidi µ-dishes (Thistle Scientific) for microinjection experiments. Dissociated cultures were maintained as explant cultures, except that B27 was replaced with 10 % fetal bovine serum (Merck) and 2.5 S NGF was lowered to 30 ng/ml. Culture media was replenished every 3 days. For most experiments, neurites were allowed to extend for 7 days before treatment.

## Drug treatments

For most experiments, DRG and SCG neurons were treated at day in vitro (DIV) 7 with vacor (Greyhound Chromatography) or vehicle ($H_2O$ with 4 % 1 N HCl), and VMN or vehicle ($H_2O$) just prior to imaging (time 0 hr). When used, FK866 (kind gift of Prof Armando Genazzani, University of Novara) and NAM (Merck) were added at the same time as vacor. For neurite outgrowth and long-term survival assays, multiple doses of vacor or vehicle were added by replacing media with fresh media containing the drugs at the timepoints indicated in the figure. The drug concentrations used are indicated in the figures and figure legends. Vacor was dissolved in $H_2O$ with 4 % 1 N HCl or DMSO; quantitation of the dissolved stock was performed spectrophotometrically ($\varepsilon_{340nm}$ 17.8 mM$^{-1}$cm$^{-1}$).

## Acquisition of phase contrast images and quantification of neurite degeneration and outgrowth

Phase contrast images were acquired on a DMi8 upright fluorescence microscope (Leica microsystems) coupled to a monochrome digital camera (Hamamatsu C4742-95). The objectives used were NPLAN 5 X/0.12 for neurite outgrowth assays and HCXPL 20 X/0.40 CORR for neurite degeneration assays. Radial outgrowth was determined by taking the average of two measurements of representative neurite outgrowth for each ganglion at DIV2-3-5-7. Measurements were made from overlapping images of the total neurite outgrowth. For neurite degeneration assays, the degeneration index was determined using a Fiji plugin (*Sasaki et al., 2009*). For each experiment, the average was calculated from three fields per condition; the total number of experiments is indicated in the figure legends.

## Microinjection and quantification of % of healthy neurites and viable neurons

DIV5 dissociated *Sarm1*-/- SCG neurons were microinjected using a Zeiss Axiovert S100 microscope with an Eppendorf FemtoJet microinjector and Eppendorf TransferMan micromanipulator. Plasmids were diluted in 0.5 X PBS (without $CaCl_2$ and $MgCl_2$) and filtered using a Spin-X filter (Costar). The mix was injected directly into the nuclei of SCG neurons using Eppendorf Femtotips (*Gilley and Loreto, 2020*). Injected plasmids were allowed to express for 2 days before vacor or vehicle treatment and axotomy. Plasmids were injected at the following concentrations: 2.5 (*Figure 2G–I*) or 10

(*Figure 4E and F*) ng/µl (untagged) wild-type hSARM1 and 2.5 ng/µl E642A hSARM1 (pCMV-Tag2 vector backbone), 10 ng/µl (C-terminal Flag-tagged) wild-type, K193R, R157A and W103A hSARM1 (pLVX-IRES-ZsGreen vector backbone) (*Figure 7A–F*), 30 ng/µl EGFP-NMN deamidase (pEGFP-C1 vector backbone), 30 ng/µl pEGFP-C1, 40 (*Figure 2G–I*; *Figure 7A–F*) and 70 (*Figure 4E and F*) ng/µl pDsRed2-N1. To check for expression of wild-type and E642A hSARM1 constructs (pCMV-Tag2 vector backbone) (*Figure 2—figure supplement 1H*), wild-type, K193R, R157A, and W103A hSARM1 (pLVX-IRES-ZsGreen vector backbone) (*Figure 7—figure supplement 1A*) and pDsRed2-N1 were injected at 25 ng/µl; neurons were then fixed in 4 % paraformaldehyde (PFA) (Merck) and immunostained with mouse monoclonal anti-SARM1 primary antibody (*Chen et al., 2011*) followed by Alexa Fluor 488 or 647 anti-mouse secondary antibody (Thermo Fisher Scientific). Fluorescence microscopy images were acquired on a DMi8 upright fluorescence microscope (Leica microsystems) coupled to a monochrome digital camera (Hamamatsu C4742-95). The objective used was HCXPL 20 X/0.40 CORR. Numbers of morphologically normal and continuous DsRed labelled neurites and morphologically normal cell bodies were counted in the same field at the indicated timepoints after vacor or vehicle treatment. For each experiment, the average was calculated from three fields per condition; the total number of experiments is indicated in the figure legends. The percentage of healthy neurites and viable neurons remaining relative to the first time point was determined.

## Microfluidic cultures

Dissociated DRG neurons were plated in microfluidic chambers (150 µm barrier, XONA Microfluidics). Cell suspension was pipetted into each side of the upper channel of the microfluidic device. On DIV7, a difference of 100 µl of media between chambers was introduced and drugs were added to the compartment with the lower hydrostatic pressure. To calculate the % of viable neurons, 1 µg/ml propidium iodide (PI) (Thermo Fisher Scientific) was added to the media 15 min before drug addition. Phase contrast and fluorescence microscopy images were acquired on a DMi8 upright fluorescence microscope (Leica microsystems) coupled to a monochrome digital camera (Hamamatsu C4742-95). The objective used was HCXPL 20 X/0.40 CORR. For each experiment, the degeneration index was calculated from the average of two distal fields of neurites per condition, whereas the % of viable neurons remaining relative to the first time point was calculated from the average of three fields of cell bodies (staining positive for PI) per condition; the total number of experiments is indicated in the figure legends.

## Intravitreal (IVT) injection into the eye

9- to 12-week-old male and female C57BL/6J (Charles River, UK, RRID:IMSR_JAX:000664), or *Sarm1*$^{-/-}$ (RRID:MGI:3765957) mice were intravitreally injected with either 2 µL vacor, DMSO (vehicle) or PBS. Injections were performed as previously described (*Osborne et al., 2018*). ERG responses were recorded simultaneously from both eyes using an Espion E3 system with full-field Ganzfeld sphere (Diagnosys, Cambridge, UK). Animals were dark-adapted overnight, and ERG recordings performed under low level, red light illumination. Peak RGC responses (pSTR) were recorded between 70–110 ms after a light exposure of –4.73 log cd.s.m$^{-2}$, B wave peaks between 80–120 ms, at –1.90 log cd.s.m$^{-2}$, and A wave troughs within the first 20 ms at 1.29 log cd.s.m$^{-2}$. Following culling via exposure to $CO_2$, eyes were post-fixed for 24 hr in 4 % PFA before retinal flatmounts were prepared as previously described (*Osborne et al., 2018*). Briefly, retinas were excised from the eye cup, flattened and stained with the RGC specific nuclei marker BRN3A (Santa Cruz, sc-31984, RRID:AB_2167511, 1:200). Images were captured, blindly, from eight regions per retina using a 20 X objective, and automatically quantified with the Fiji plugin 'Simple RGC' (*Cross et al., 2021*). The representative images shown were acquired on a confocal microscope (Leica Microsystems) using a 40 X objective.

## Determination of NMN, NAD, cADPR, vacor, VMN, and VAD tissue levels

Following treatment with vacor or vehicle, DIV7 wild-type and *Sarm1*$^{-/-}$ DRG ganglia were separated from their neurites with a scalpel. Neurite and ganglia (containing short proximal neurite stumps as well as cell bodies) fractions were washed in ice-cold PBS and rapidly frozen in dry ice and stored at –80 °C until processed. Tissues were ground in liquid $N_2$ and extracted in $HClO_4$ by sonication, followed by neutralisation with $K_2CO_3$. NMN and NAD were subsequently analysed by spectrofluorometric HPLC

analysis after derivatisation with acetophenone (*Mori et al., 2014*). Vacor, VMN, and VAD were determined in DRG whole explant cultures by ion pair C18-HPLC chromatography, as previously described (*Buonvicino et al., 2018*). cADPR levels were determined in DRG whole explant cultures using a cycling assay, as previously described (*Graeff and Lee, 2002*). A single analysis of vacor, VMN, VAD, and cADPR levels was performed in DRG neurite and ganglia fractions independently, which was not further repeated due to the low basal levels of these metabolites and the amount of cellular material needed for this type of analysis. Metabolite levels were normalised to protein levels quantified with the Bio-Rad Protein Assay (Bio-Rad) on formate-resuspended pellets from the aforementioned $HClO_4$ extraction.

## VMN and VAD synthesis and purification

VMN and VAD were synthesised as previously reported (*Buonvicino et al., 2018*) with minor changes. Vacor was either phosphoribosylated in vitro by murine NAMPT (mNAMPT) into VMN or phosphoribosylated by mNAMPT and adenylated by murine NMNAT2 (mNMNAT2) into VAD. The scheme of these reactions and the reaction mixtures are shown in *Figure 5—figure supplement 2A,B*. Inorganic yeast pyrophosphatase (PPase), phosphoribosyl 1-pyrophosphate (PRPP) and adenosine triphosphate (ATP) were all from Merck. Recombinant mNAMPT and mNMNAT2 were purified as previously described (*Amici et al., 2017*; *Orsomando et al., 2012*). Following incubation, reaction mixtures (A) for VMN and (A + B) for VAD (*Figure 5—figure supplement 2B*) were stopped by rapid cooling on ice and kept refrigerated until injection for purification. Next, the VMN and VAD obtained were purified by FPLC under volatile solvents and lyophilised. Briefly, a preparative IEC chromatography was carried out on AKTA Purifier onto the anion exchanger resin Source 15Q (GE HealthCare, 20 ml volume). The column was equilibrated at room temperature (~25 °C), at 5 ml/min. After injection of the two mixtures above, a linear gradient elution was applied by mixing the two volatile buffers, as indicated. The eluate was monitored at wavelengths of 260 nm and 340 nm (optimal for vacor nucleotides), and both VMN and VAD peaks were collected (*Figure 5—figure supplement 2C,D*). Next, an additional chromatography was performed by HPLC onto a RP C18 column (Varian, 250 × 4.6mm, 5μm particles) heated at 50 °C to obtain a pure VAD stock. The equilibration in such case was carried out at 1 ml/min in ammonium formate buffer; this was followed by multiple injections of the previously collected IEC pool of VAD (~1 ml each), and elution by a linear gradient of increasing acetonitrile in the buffer. The eluate was monitored again at wavelengths 260 nm and 340 nm and the VAD peak was collected (*Figure 5—figure supplement 2D*). After lyophilisation, dry pellets were stored at –80 °C. The resulting VMN and VAD lyophilised powders were 100 % pure. Their quantitation after resuspension in $H_2O$ was performed spectrophotometrically ($\varepsilon_{340nm}$ of 17.8 $mM^{-1}cm^{-1}$).

## Recombinant hSARM1 purification

Human embryonic kidney (HEK) 293T cells (clone 17, [HEK 293T/17]) were obtained from the American Type Culture Collection (ATCC, CRL-11268, RRID:CVCL_1926) and had been authenticated by STR profiling. Mycoplasma contamination was not detected. Recombinant, full-length C-terminal Flag-tagged wild-type and K193R human SARM1 (hSARM1-Flag) were expressed in HEK293T cells and purified by immunoprecipitation. HEK cells at 50–70% confluence were transfected with wild-type and K193R hSARM1-Flag expression construct (pLVX-IRES-ZsGreen vector backbone) using Lipofecatmine 2000 (Thermo Fisher Scientific). To boost wild-type and K193R hSARM1-Flag expression, media were supplemented with 2 mM nicotinamide riboside (prepared from Tru Niagen capsules by dissolving the contents and passing through a 0.22 μm filter) at the time of transfection. After 24 hr, cells were collected and washed in cold PBS and lysed for 10 min with trituration by pipetting and repeated vortexing in ice-cold KHM buffer (110 mM potassium acetate, 20 mM HEPES pH 7.4, 2 mM $MgCl_2$, 0.1 mM digitonin) with cOmplete, Mini, EDTA-free protease inhibitor cocktail (Roche). After centrifugation for 5 min at 3000 rpm in a chilled microfuge, the supernatant was collected and diluted to 1 μg/μl in KHM buffer after determination of its protein concentration by the Pierce BCA assay (Thermo Fisher Scientific). For immunoprecipitation, 1 ml of extract was incubated overnight at 4 °C with rotation with 20 μg/ml anti-FLAG M2 antibody (Merck, F3165, RRID:AB_259529) and 50 μl Pierce magnetic protein A/G beads (Thermo Fisher Scientific) pre-washed with KHM buffer. Beads were collected on a magnetic rack and washed 3 x with 500 μl KHM buffer and 1 x with PBS (with protease inhibitors) and then resuspended in PBS containing 1 mg/ml BSA. hSARM1-Flag concentration in

the bead suspension was determined relative to an hSARM1 standard (purified from insect cells) by immunoblotting using a rabbit polyclonal antibody raised against the SAM domains of human SARM1 (both kindly provided by AstraZeneca).

## hSARM1 NADase activity

Rates of NAD consumption by recombinant wild-type and K193R hSARM1 were measured by HPLC under a discontinuous assay that was set as follows. Typically, mixtures of 0.02–0.2 ml contained 2–20 µg/ml of hSARM1-Flag (on-beads) in buffer HEPES/NaOH 50 mM, pH 7.5, and 250 µM NAD. NMN, VMN, and VAD were added at the concentrations indicated in the figures. Reactions were initiated by adding NAD, incubated at 25 °C in a water-bath, stopped at appropriate times by $HClO_4$ treatment, neutralised with $K_2CO_3$, and subsequently analysed by ion-pair reverse-phase (RP) HPLC (NAM, NMN, NAD, ADPR, cADPR) (*Mori et al., 2014*) or under optimised conditions for VMN and VAD detection (*Buonvicino et al., 2018*). The products formed (NAM, ADPR, and cADPR) were quantified from the area of separated peaks. Rates were calculated under NAD consumption ≤20 % from the linearly accumulating products ADPR and cADPR (*Figure 5—figure supplement 1B*). One unit (U) of activity represents the enzyme amount that forms 1 µmol/min of the products above under these assay conditions. Data from three independent experiments were averaged to calculate hSARM1 activity fold activation (*Figure 5A*) and then studied by fitting through the Excel software package to the rate equation below

$$V = \frac{V_{max}\left(X + \left(\frac{1-X}{1+\left(\frac{[I]}{K_a}\right)}\right)\right)\left(Y + \left(\frac{1-Y}{1+\frac{[I]}{k_i}^n}\right)\right)[S]}{K_m + [S]}$$

that represents a Michaelis-Menten equation adapted to be valid for readily reversible effectors, that is, including activators and inhibitors binding via distinct sites to one protein target. Maximum velocity ($V_{max}$) represents the rate in the absence of the effector (basal activity) corrected by the two parameters between parentheses, the first for activation and the second for inhibition, taking into account opposite and independent contributions exerted by the single but dual effector I. $X$ is the 'relative activity fold activation' factor at saturating concentrations of I, $Y$ is the residual activity fraction at saturating concentration of I, $K_a$ is the affinity constant of I for the activation site, $K_i$ is the affinity constant of I for the inhibition site, and $n$ is the Hill coefficient indicating cooperativity (if ≠ 1).

## Expression and purification of recombinant dSARM1[ARM]

Codon-optimised cDNA of dSARM1[ARM] (residues 315–678) was cloned into the pMCSG7 expression vector at the SspI site, using ligation-independent cloning (*Aslanidis and de Jong, 1990*). Plasmids encoding various dSARM1[ARM] mutants were generated using the Q5 Site-Direct Mutagenesis Kit (New England BioLabs). Plasmids were transformed into BL21 (DE3) cells. Large-scale expression of dSAR-M1[ARM] and its mutants was performed using the auto-induction method (*Studier, 2005*). In brief, 2 ml of overnight culture of the transformed BL21 cells was inoculated into 1 l of auto-induction media with 100 µg/ml ampicillin, and grew at 37 °C, 225 rpm to reach $OD_{600}$ of 0.8–1.0. Temperature was decreased to 20 °C for protein expression overnight. Cells were harvested by centrifugation at 6000× g for 20 min at 4 °C, and resuspended into lysis buffer (50 mM HEPES pH 8.0, 500 mM NaCl and 30 mM imidazole). PMSF (phenylmethanesulfony fluoride) was added to the cell resuspension at 1 mM concentration. Cell resuspension from 1 l of culture was sonicated at the amplitude of 40 % for 120 s (10 s on and 20 s off). Lysed cells were centrifuged at 15300× g for 40 min at 4 °C. The supernatant was loaded onto the HisTrap HP 5 ml Ni column, followed by washing using 10 column volumes of lysis buffer. The bound protein was eluted using elution buffer (50 mM HEPES pH 8.0, 500 mM NaCl and 300 mM imidazole), and incubated with TEV (tobacco etch virus) protease to remove the N-terminal 6× histidine tag in the SnakeSkin Dialysis Tubing, 3.5 K MWCO (Thermo Fisher Scientific), dialysed against the buffer containing 20 mM HEPES (pH 8.0), 300 mM NaCl and 1 mM DTT at 4 °C overnight. The cleaved protein was reloaded onto the Ni column and flow-through was collected, concentrated to 10 ml and injected onto the Superdex 75 HiLoad 26/600 column (GE Healthcare) equilibrated with gel-filtration buffer (10 mM HEPES pH 8.0 and 150 mM NaCl). The pure target protein, as confirmed by SDS-PAGE analysis and mass spectrometry, was concentrated to 10 mg/ml and stored at –80 °C.

Protein crystallisation, diffraction data collection and structural determination dSARM1$^{ARM}$ (10 mg/ml) and VMN were incubated at 1: 10 molar ratio at 4 °C overnight. Hanging drops, containing 2 μl of protein complex and 2 μl of well solution (20 % PEG 3350 and 0.2 M magnesium acetate tetrahydrate, pH 7.9), were equilibrated against 500 μl of well solution at 20 °C. Crystals with irregular shapes were observed after 3–5 days. In situ partial proteolysis happened during the process of crystallisation, as the analysis of the crystals via SDS-PAGE analysis and mass spectrometry indicated that the crystallised protein only spanned residues 370–678, rather than residues 315–678. Crystals cryo-protected in paratone-N diffracted to 1.7–2.3 Å resolution at the Australian Synchrotron MX2 beamline. X-ray diffraction data was collected at the wavelength of 0.95372 Å. Raw data was processed using XDS (RRID:SCR_015652) (*Kabsch, 2010*); data reduction, scaling and reflection selection for R$_{free}$ calculation were performed using Aimless within CCP4 suite (RRID:SCR_007255) (*Evans and Murshudov, 2013*). Phases were calculated via molecular replacement using the NMN-bound dSARM1$^{ARM}$ (PDB: 7LCZ) (*Figley et al., 2021*; *Gu et al., 2021*) as a search model in the program Phaser within Phenix (RRID:SCR_014224) (*McCoy et al., 2007*). The model was refined using COOT (RRID:SCR_014222) (*Emsley and Cowtan, 2004*) and Phenix (*Afonine et al., 2012*), and analysed using PyMol (RRID:SCR_000305) (Schrodinger), PDBsum (RRID:SCR_006511) (*Laskowski et al., 2018*) and PISA (RRID:SCR_015749) (*Krissinel and Henrick, 2007*). The crystals contain two molecules of VMN-bound dSARM1$^{ARM}$ (RMSD = 0.3 Å over 296 Cα atoms) in the asymmetric unit.

## Isothermal titration calorimetry (ITC)

ITC experiments were performed in duplicate on Nano ITC (TA Instruments). All proteins and compounds were dissolved in a buffer containing 10 mM HEPES (pH 8.0) and 150 mM NaCl. The baseline was equilibrated for 600 s before the first injection. VMN (0.6 mM) was titrated as 20–25 injections of 1.96 μL every 200 s, into 45 μM protein. The heat change was recorded by injection over time and the binding isotherms were generated as a function of molar ratio of the protein solution. The dissociation constants (K$_d$) were obtained after fitting the integrated and normalised data to a single-site binding model using NanoAnalyze (TA Instruments).

## NMN deamidase activity

Recombinant *E. coli* NMN deamidase was obtained as described previously (*Zamporlini et al., 2014*). Activity was measured in buffer HEPES/NaOH 50 mM, pH 7.5, in the presence of 4 milliU/ml enzyme, 0.5 mg/ml BSA, and 250 μM NMN or VMN. Vacor, VMN, and VAD, all at the concentration of 250 μM, were also assayed in presence of the substrate NMN. Reactions were incubated at 37 °C, then stopped and analysed by HPLC using the two methods described above (*Buonvicino et al., 2018*; *Mori et al., 2014*). The NMN deamidase rates were calculated after separation and quantification of the NaMN product formed from NMN, and finally reported as relative percentages of controls in the presence of NMN alone.

## Western blot

Following treatment with vacor, DRG ganglia were separated from their neurites with a scalpel. Neurites and ganglia were collected, washed in ice-cold PBS containing protease inhibitors (Roche), and lysed directly in 15 μl 2 x Laemmli buffer containing 10 % 2-mercaptoethanol (Merck). Samples were loaded on a 4-to-20% SDS polyacrylamide gel (Bio-Rad). Membranes were blocked for 3 hr in 5 % milk in TBS (50 mM Trizma base and 150 mM NaCl, PH 8.3, both Merck) plus 0.05 % Tween-20 (Merck) (TBST), incubated overnight with primary antibody in 5 % milk in TBST at 4 °C and subsequently washed in TBST and incubated for 1 hr at room temperature with HRP-linked secondary antibody (Bio-Rad) in 5 % milk in TBST. Membranes were washed, treated with SuperSignal West Dura Extended Duration Substrate (Thermo Fisher Scientific) and imaged with Uvitec Alliance imaging system. The following primary antibodies were used: mouse monoclonal anti-SARM1 (*Chen et al., 2011*) (1:5000) and mouse anti-β-actin (Merck, A5316, RRID:AB_476743, 1:2000) as a loading control. Quantification of band intensity was determined by densitometry using Fiji (RRID:SCR_002285).

## Statistical analysis

Statistical testing of data was performed using Prism (GraphPad Software, La Jolla, USA, RRID:SCR_002798). The appropriate tests used and the n numbers of each individual experiment are described in the figure legends. A p-value < 0.05 was considered significant.

## Acknowledgements

We thank members of the Coleman lab, Kobe lab, Prof James Fawcett and Prof Nadia Raffaelli for useful discussions. We also thank Dr Lucia Silvestrini for help in obtaining the *Neurospora crassa* NADase for the cADPR detection assay. We acknowledge the use of the University of Queensland Remote Operation Crystallization and X-ray (UQROCX) Facility at the Centre for Microscopy and Microanalysis and the Australian Synchrotron MX beamlines, and the staff for support.

## Additional information

### Competing interests

Weixi Gu: Weixi Gu receives research funding from Disarm Therapeutics, a wholly-owned subsidiary of Eli Lilly & Co, Cambridge, MA, USA, but they had no role in the research presented here. Andrew Osborne: Andrew Osborne is affiliated with Ikarovec Ltd. The author has no financial interests to declare. Zhenyao Luo: Zhenyao Luo receives research funding from Disarm Therapeutics, a wholly-owned subsidiary of Eli Lilly & Co, Cambridge, MA, USA, but they had no role in the research presented here. Thomas Ve: Thomas Ve receives research funding from Disarm Therapeutics, a wholly-owned subsidiary of Eli Lilly & Co, Cambridge, MA, USA, but they had no role in the research presented here. Laura M Desrochers: This work is in part funded by a BBSRC/AstraZeneca Industrial Partnership Award and Laura M Desrochers was an employee of AstraZeneca for part of the project. Laura M Desrochers is affiliated with Vertex Pharmaceuticals. The author has no financial interests to declare. Qi Wang: This work is in part funded by a BBSRC/AstraZeneca Industrial Partnership Award and Qi Wang was an employee of AstraZeneca for part of the project. Qi Wang is affiliated with Kymera Therapeutics. The author has no financial interests to declare. Bostjan Kobe: Bostjan Kobe is a consultant and shareholder of Disarm Therapeutics and receives research funding from Disarm Therapeutics, a wholly-owned subsidiary of Eli Lilly & Co, Cambridge, MA, USA, but they had no role in the research presented here. Michael P Coleman: Michael P Coleman holds funding jointly provided by AstraZeneca for academic research and consults for Nura Bio, neither of which had a role in the research presented here. The other authors declare that no competing interests exist.

### Funding

| Funder | Grant reference number | Author |
| --- | --- | --- |
| Wellcome Trust | 210904/Z/18/Z | Andrea Loreto |
| Wellcome Trust | 206634 | Peter Arthur-Farraj |
| Biotechnology and Biological Sciences Research Council | BB/S009582/1 | Andrea Loreto Jonathan Gilley Michael P Coleman |
| Università Politecnica delle Marche | RSA 2016-18 and 2017-19 | Giuseppe Orsomando |
| National Health and Medical Research Council | NHMRC 1160570 | Bostjan Kobe Thomas Ve |
| Sight Research UK | SAC 041 | Andrew Osborne Bart Nieuwenhuis |
| Australian Research Council | FL180100109 | Bostjan Kobe |
| Australian Research Council | FT200100572 | Thomas Ve |

| Funder | Grant reference number | Author |
|---|---|---|
| National Health and Medical Research Council | NHMRC 1196590 | Thomas Ve |

The funders had no role in study design, data collection and interpretation, or the decision to submit the work for publication.

## Author contributions

Andrea Loreto, Conceptualization, Data curation, Formal analysis, Funding acquisition, Investigation, Methodology, Project administration, Supervision, Validation, Visualization, Writing – original draft, Writing – review and editing; Carlo Angeletti, Formal analysis, Investigation, Methodology; Weixi Gu, Formal analysis, Investigation, Writing – original draft, Writing – review and editing; Andrew Osborne, Formal analysis, Investigation, Writing – review and editing; Bart Nieuwenhuis, Elisa Merlini, Lauren Hartley-Tassell, Thomas Ve, Investigation; Jonathan Gilley, Investigation, Methodology, Writing – review and editing; Peter Arthur-Farraj, Investigation, Writing – review and editing; Adolfo Amici, Formal analysis; Zhenyao Luo, Formal analysis, Investigation; Laura M Desrochers, Qi Wang, Methodology; Bostjan Kobe, Formal analysis, Investigation, Supervision, Writing – original draft, Writing – review and editing; Giuseppe Orsomando, Conceptualization, Formal analysis, Funding acquisition, Investigation, Methodology, Supervision, Validation, Writing – review and editing, Writing – original draft; Michael P Coleman, Conceptualization, Funding acquisition, Project administration, Supervision, Validation, Writing – original draft, Writing – review and editing

## Author ORCIDs

Andrea Loreto http://orcid.org/0000-0001-6535-6436
Weixi Gu http://orcid.org/0000-0002-1185-3557
Bart Nieuwenhuis http://orcid.org/0000-0002-2065-2271
Jonathan Gilley http://orcid.org/0000-0002-9510-7956
Peter Arthur-Farraj http://orcid.org/0000-0002-1239-9392
Adolfo Amici http://orcid.org/0000-0002-1081-7749
Bostjan Kobe http://orcid.org/0000-0001-9413-9166
Giuseppe Orsomando http://orcid.org/0000-0001-6640-097X

## Ethics

All studies conformed to the institution's ethical requirements in accordance with the 1986 Animals (Scientific Procedures) Act under Project Licences PPL P98A03BF9 and PP1824519, and in accordance with the Association for Research in Vision and Ophthalmology's Statement for the Use of Animals in Ophthalmic and Visual Research.

## Decision letter and Author response

Decision letter https://doi.org/10.7554/eLife.72823.sa1
Author response https://doi.org/10.7554/eLife.72823.sa2

# Additional files

## Supplementary files
• Supplementary file 1. X-ray data collection and structural refinement statistics.
• Transparent reporting form

## Data availability

All data generated or analysed during this study are included in the manuscript and supporting file; Source Data files have been provided for Figures 1-7 and figure supplements. VMN-bound dSAR-M1ARM crystal structure has been deposited in the Protein Data Bank (PDB: 7M6K).

The following dataset was generated:

| Author(s) | Year | Dataset title | Dataset URL | Database and Identifier |
|---|---|---|---|---|
| Loreto A | 2021 | VMN-bound dSARM1ARM crystal structure | https://www.rcsb.org/structure/7M6K | RCSB Protein Data Bank, 7M6K |

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
