## [Editor Report]

Axon degeneration is activated by injury and through a pathway that involves the SARM1 protein, which possesses NAD^+^ cleavage activity. This manuscript definitively identifies the pesticide vacor and its metabolite VMN as an activator of Sarm1. The study works out the mechanism of activation via structural analysis of the allosteric binding site. Because the axon degeneration pathway is activated in a number of neurodegenerative contexts, the insights into the mechanism of action of vacor during neurotoxicity provide avenues for future therapeutic strategies.

---

## [Decision Letter]

**Decision letter after peer review:**

Thank you for submitting your article "Neurotoxin-mediated potent activation of the axon degeneration regulator SARM1" for consideration by *eLife*. Your article has been reviewed by 3 peer reviewers, and the evaluation has been overseen by a Reviewing Editor and Suzanne Pfeffer as the Senior Editor. The following individuals involved in review of your submission have agreed to reveal their identity: Marc Freeman (Reviewer #2); Mohanish Deshmukh (Reviewer #3).

The reviewers have discussed their reviews with one another, in consultation with the Reviewing Editor. Among the comments, the overall enthusiasm is high and the results of vacor and its metabolite VMN as an activator of Sarm were considered to be well done and novel. The significance of the results and the structural data was also strong. Several revisions were recommended to improve the clarity and to extend the findings.

Essential revisions:

1) VMN is an activator of Sarm at low concentration and an inhibitor at high concentrations. Is VMN's inhibitory activity working on the active site? Is there any additional evidence for how VMN inhibits Sarm either through the active site or through an allosteric site. Two of the three reviewers requested further insight into this question and felt the answer would give mechanistic insight into a new activator and inhibitor of SARM.

2) Do high VMN concentrations (or vacor) inhibit axotomy induced axon degeneration? Any data to address this question should be included.

3) Some reorganization of the manuscript would be helpful. Many of the supplemental figures contains key data. They should be included with the four figures presently in the submission. The Introduction is dated and does not include consideration of the allosteric pocket.

*Reviewer #1 (Recommendations for the authors):*

1) Is the supplement to figure 1 the authors use compartment chambers to show that vacor can trigger local degeneration of cell bodies and axons. They state that with cell body application "vacor-induced cell death causes eventual secondary neurite degeneration." However, in their data, cell body application of vacor causes more rapid and robust distal axon degeneration than does direct application of vacor to the distal axons. This is not consistent with the "eventual neurite degeneration" and suggests a different and more active mechanism of degeneration in that case. Is the axon degeneration in the distal compartment still Sarm dependent? This should at a minimum be discussed, and now that inhibitors of Sarm are available could be experimentally addressable.

2) The authors report that VMN not only activates Sarm, but also inhibits Sarm. The authors claims that this inhibitory function "could reveal critical information on how Sarm activity is regulated," yet they do not attempt to understand the mechanism of inhibition. Is the inhibitory role of VMN working through the same allosteric pocket? Is it acting on the catalytic site in the enzyme? With the nice biochemical system the authors have established this seems easily addressable and would be an important addition to a manuscript describing how VMN regulates Sarm activity.

3) The introduction in this manuscript is quite similar to the version published on Bioarchive about a year ago. In that year, there has been tremendous progress in understanding the mechanism of Sarm activation, with the publication of four papers describing the structure of allosteric ligands binding to Sarm and the definition of the roles of NMN and NAD in Sarm activation/inhibition. The introduction should be updated in order to better place the current work in context.

4) There are only four figures but many supplemental figures that are quite important. The paper would be much easier to follow if it were re-organized into a more traditional 7-8 figure format.

5) The final sentence in the abstract claims that the study shows that Sarm can "permanently block programmed axon death induced by toxicity." This is overstated, as the authors only show Sarm knockout protects for three days after a single vacor injection, and in vitro after two doses separated by two days. This does not speak to whether the protection is permanent, since this would require studies of long-term toxin exposure.

6) The manuscript would benefit from a more comprehensive analysis of how this work relates to prior work published by these authors. For example, the point mutants in the binding pocket of Sarm were previously analyzed by these authors for their role in NMN binding and Sarm activation. This should be mentioned and the findings compared. Similarly, there is a comparison of the crystal structure for NMN and NAD, but not for their recently described structure of Sarm bound to NAMN.

7) The authors show very convincingly that Sarm activation leads to loss of retinal ganglion cells. A couple of previous studies using different injury models have come to different conclusions about whether Sarm causes RGC death. It would be interesting to comment on how the author's current data relates to those prior findings in Sarm knockouts.

8) The abstract claims that the structure of VMN bound to the Sarm regulatory domain will "facilitate drug development for some human disorders including ALS and some polyneuropathies." Since this binding pocket has already been well defined in previous studies, these is too grand of a claim for the abstract, which should focus more on what the manuscript has actually demonstrated. Commentary on hopes for improved drug development would be more appropriate for the Discussion section.*Reviewer #2 (Recommendations for the authors):*

None. This is a rigorous and complete study.

*Reviewer #3 (Recommendations for the authors):*

The experiments very well done and the results are interesting. The use of multiple models both in vitro and in vivo as well as the biochemical and structural studies are a strength of this study.

---

## [Author Response]

Essential revisions:1) VMN is an activator of Sarm at low concentration and an inhibitor at high concentrations. Is VMN's inhibitory activity working on the active site? Is there any additional evidence for how VMN inhibits Sarm either through the active site or through an allosteric site. Two of the three reviewers requested further insight into this question and felt the answer would give mechanistic insight into a new activator and inhibitor of SARM.

We appreciate these comments. We agree that VMN-mediated inhibition of SARM1 is interesting and understanding how this inhibition works is certainly one of our future plans. We have started working towards this and our preliminary data suggest that the mechanism of inhibition might not simply be explained by VMN binding to the active or allosteric site. Therefore, unravelling how VMN inhibits SARM1 is likely to be a fairly substantial undertaking at this stage, which we feel goes beyond the scope of the current study. We prefer to keep the focus at present on the toxic effect of vacor and its metabolite VMN, along with the many exciting implications of this including drug development and the potential for activation of SARM1 by other environmental toxins.

2) Do high VMN concentrations (or vacor) inhibit axotomy induced axon degeneration? Any data to address this question should be included.

We thank the reviewers for these interesting suggestions; there are, however, some experimental limitations which need to be considered. For instance, vacor is insoluble in culture media at concentrations higher than 100µM, limiting the range of concentrations that can be tested. We have treated axotomised neurites with 100µM vacor; as expected, we did not see protection as vacor is normally very toxic at this concentration.

Similarly, it is unclear how much VMN actually enters the cell, considering that it is a charged molecule. 500µM VMN (the concentration used in this study) is the first concentration that causes toxicity in neurons. This is substantially higher than the first toxic concentration of vacor (50µM), suggesting that VMN cell-permeability is limited. Therefore, reaching high enough concentrations of VMN in neurons to confer protection is challenging.

We would be happy to share preliminary data on these points for revision purposes, but we would prefer not to add these to the manuscript as interpretation of results in this instance is subject to many caveats.

3) Some reorganization of the manuscript would be helpful. Many of the supplemental figures contains key data. They should be included with the four figures presently in the submission. The Introduction is dated and does not include consideration of the allosteric pocket.

We appreciate this feedback and have made the following changes to address reviewers’ comments:

– We have changed the format of the paper into a more traditional 7 figure paper. Some important data that were in supplementary have now been moved to the main text. Specifically, Figure 1 —figure supplement 1 is now Figure 1. We have added a comparison with the structure of SARM1 bound to NaMN. All structural data are now in Figure 6. Finally, we have moved all data on SARM1 mutants in figure 7.

– We have added a paragraph in the introduction to include the latest work in the field; [page 4, lines 62-70].

– We have made a number on minor changes in response to reviewers’ comments, detailed below. We have used the word tracked changes function to highlight changes to the text.

Reviewer #1 (Recommendations for the authors):1) Is the supplement to figure 1 the authors use compartment chambers to show that vacor can trigger local degeneration of cell bodies and axons. They state that with cell body application "vacor-induced cell death causes eventual secondary neurite degeneration." However, in their data, cell body application of vacor causes more rapid and robust distal axon degeneration than does direct application of vacor to the distal axons. This is not consistent with the "eventual neurite degeneration" and suggests a different and more active mechanism of degeneration in that case. Is the axon degeneration in the distal compartment still Sarm dependent? This should at a minimum be discussed, and now that inhibitors of Sarm are available could be experimentally addressable.

We thank the reviewer for pointing this out. We have discussed this important point in the main text; [page 7; lines 129-135].

2) The authors report that VMN not only activates Sarm, but also inhibits Sarm. The authors claims that this inhibitory function "could reveal critical information on how Sarm activity is regulated," yet they do not attempt to understand the mechanism of inhibition. Is the inhibitory role of VMN working through the same allosteric pocket? Is it acting on the catalytic site in the enzyme? With the nice biochemical system the authors have established this seems easily addressable and would be an important addition to a manuscript describing how VMN regulates Sarm activity.

We appreciate these comments. We agree that VMN-mediated inhibition of SARM1 is interesting and understanding how this inhibition works is certainly one of our future plans. We have started working towards this and our preliminary data suggest that the mechanism of inhibition might not simply be explained by VMN binding to the active or allosteric site. Therefore, unravelling how VMN inhibits SARM1 is likely to be a fairly substantial undertaking at this stage, which we feel goes beyond the scope of the current study. We prefer to keep the focus at present on the toxic effect of vacor and its metabolite VMN, along with the many exciting implications of this including drug development and the potential for activation of SARM1 by other environmental toxins.

3) The introduction in this manuscript is quite similar to the version published on Bioarchive about a year ago. In that year, there has been tremendous progress in understanding the mechanism of Sarm activation, with the publication of four papers describing the structure of allosteric ligands binding to Sarm and the definition of the roles of NMN and NAD in Sarm activation/inhibition. The introduction should be updated in order to better place the current work in context.

We thank the reviewer for the suggestion. We have now updated the introduction to reflect recent developments in the field; [page 4, lines 62-70].

4) There are only four figures but many supplemental figures that are quite important. The paper would be much easier to follow if it were re-organized into a more traditional 7-8 figure format.

We have now reorganised the manuscript into a more traditional 7 figure format. Specifically, Figure 1 —figure supplement 1 is now Figure 1. All structural data are now in Figure 6. Finally, we have moved all data on SARM1 mutants which were previously in supplementary material in figure 7.

5) The final sentence in the abstract claims that the study shows that Sarm can "permanently block programmed axon death induced by toxicity." This is overstated, as the authors only show Sarm knockout protects for three days after a single vacor injection, and in vitro after two doses separated by two days. This does not speak to whether the protection is permanent, since this would require studies of long-term toxin exposure.

We thank the reviewer for pointing this out. We have changed the word “permanently” with “robustly” in the abstract.

6) The manuscript would benefit from a more comprehensive analysis of how this work relates to prior work published by these authors. For example, the point mutants in the binding pocket of Sarm were previously analyzed by these authors for their role in NMN binding and Sarm activation. This should be mentioned and the findings compared. Similarly, there is a comparison of the crystal structure for NMN and NAD, but not for their recently described structure of Sarm bound to NAMN.

We thank the reviewer for this suggestion. We have now added the comparison with the structure of SARM1 bound to NaMN (Figure 6), and discussed this in the text; [page 12; lines 250-264].

7) The authors show very convincingly that Sarm activation leads to loss of retinal ganglion cells. A couple of previous studies using different injury models have come to different conclusions about whether Sarm causes RGC death. It would be interesting to comment on how the author's current data relates to those prior findings in Sarm knockouts.

We have now included a sentence in the discussion on this interesting point; [page 14; lines 309-317].

8) The abstract claims that the structure of VMN bound to the Sarm regulatory domain will "facilitate drug development for some human disorders including ALS and some polyneuropathies." Since this binding pocket has already been well defined in previous studies, these is too grand of a claim for the abstract, which should focus more on what the manuscript has actually demonstrated. Commentary on hopes for improved drug development would be more appropriate for the Discussion section.

We thank the reviewer for this suggestion and we have now made changes to the abstract in line with the reviewer’s suggestion; [page 3; lines 38-40].